# Larval microbiota primes the *Drosophila* adult gustatory response

Martina Montanari[1], Gérard Manière [2], Martine Berthelot-Grosjean [2], Yves Dusabyinema[3], Benjamin Gillet[3], Yaël Grosjean [2], C. Léopold Kurz[1] ✉ & Julien Royet [1] ✉

The survival of animals depends, among other things, on their ability to identify threats in their surrounding environment. Senses such as olfaction, vision and taste play an essential role in sampling their living environment, including microorganisms, some of which are potentially pathogenic. This study focuses on the mechanisms of detection of bacteria by the *Drosophila* gustatory system. We demonstrate that the peptidoglycan (PGN) that forms the cell wall of bacteria triggers an immediate feeding aversive response when detected by the gustatory system of adult flies. Although we identify ppk23+ and Gr66a+ gustatory neurons as necessary to transduce fly response to PGN, we demonstrate that they play very different roles in the process. Time-controlled functional inactivation and in vivo calcium imaging demonstrate that while ppk23+ neurons are required in the adult flies to directly transduce PGN signal, Gr66a+ neurons must be functional in larvae to allow future adults to become PGN sensitive. Furthermore, the ability of adult flies to respond to bacterial PGN is lost when they hatch from larvae reared under axenic conditions. Recolonization of germ-free larvae, but not adults, with a single bacterial species, *Lactobacillus brevis*, is sufficient to restore the ability of adults to respond to PGN. Our data demonstrate that the genetic and environmental characteristics of the larvae are essential to make the future adults competent to respond to certain sensory stimuli such as PGN.

In nature, animals live in a variety of ecological niches colonized by bacteria, viruses and fungi. As a result, animals interact with, and sometimes even host, these co-inhabitants throughout their development and during adulthood. For the better, as these microbial communities can have a positive impact on various physiological parameters of the host such as fertility, longevity and growth, to name but a few[1–7]. For the worse, as some of these microbes can negatively affect the health and homeostasis of the host[8]. The ability to detect and respond to these potentially harmful threats is an innate process fundamental to animal survival, that is conserved in all species[9,10]. To defend themselves against pathogens, animals have evolved finely tuned cellular and humoral innate immune mechanisms that preserve the physical integrity and health of the host and its offspring[11,12]. Defensive responses triggered upon microbial detection come at a cost to the host and might not always be successful[13–16]. Prior detection of potential dangers in the environment and preparations to face them could be complementary to the canonical responses elicited by pathogens within the host and represent a first line of defense[17–19]. Thus, the nervous system's perception of a microbial threat may allow the host to adopt behaviors aimed at reducing the consequences of the infestation on itself and its offspring. These behaviors can act at the individual or population level. Ants and bees have developed social

[1]Aix-Marseille Université, CNRS, IBDM, Marseille, France. [2]Centre des Sciences du Goût et de l'Alimentation, AgroSup Dijon, CNRS, INRAe, Université Bourgogne, F-21000 Dijon, France. [3]Institut de Génomique Fonctionnelle de Lyon, Ecole Normale Supérieure de Lyon, CNRS UMR5242, F-69007 Lyon, France. ✉e-mail: leopold.kurz@univ-amu.fr; julien.royet@univ-amu.fr

and behavioral immune mechanisms to defend themselves against infection through grooming[20–22]. Studies in vertebrates and invertebrates have indicated that multiple sensory systems, including olfaction, hearing and vision, are involved in detecting biological threats[18,19,23,24]. In *Drosophila*, the subject of this study, it was found that hygienic grooming behaviors can be induced by fungal molecules or bacterial contact chemicals, via different receptors and neural circuits[25,26]. Volatile chemicals, such as geosmin, released by potentially pathogenic fungi can be detected by insects via olfactory receptors and act as repellent, lowering the food intake and modulating the egg-laying rate[27]. Hallmarks of bacteria are cell wall components such as peptidoglycan (PGN) or lipopolysaccharide (LPS) that are important ligands for receptors allowing eucaryotes to detect and differentiate them from other living organisms[28]. Interestingly, these receptors are expressed on immune-competent cells as well as on neuronal cells[18,29,30]. Work from several laboratories has shown that bacterial peptidoglycan, an essential component of the bacterial cell wall, mediates many interactions between bacteria and flies[31–35]. Its recognition by PGRP family members activates NF-κB pathways on immunocompetent cells, leading to the production of immune effectors and regulators. We have recently shown that the same ligand/receptor interactions maintain a molecular dialogue between bacteria and neurons in the central and peripheral nervous systems of flies[36–38]. We now show that PGN detection by taste neurons triggers an immediate aversive response in adult flies. We identify neurons defined by the expression of the ppk23 gene (ppk23 + ) as mediators of this bacteria-induced behavior. We also demonstrate that, to be sensitive to PGN, adult flies must hatch from larvae reared in the presence of bacteria and whose Gr66a+ neuronal circuitry is functional. Thus, larval co-habitation with bacteria is a prerequisite for an adult behavioral response to PGN. This demonstrates that the genetic characteristics of the larvae as well as the environmental conditions in which they live are essential to initiate a sensory response to certain molecules later in the adult stage.

## Results

### Bacterial PGN can trigger PER in *Drosophila*

Some Peptidoglycan Recognition Protein family members are expressed in proboscis-hosted taste neurons[38,39]. One of them, the membrane receptor PGRP-LC, as well as downstream components of the IMD pathway, are functionally required for transduction of the bacterial PGN signal in these cells[31,38,40,41]. To test whether this cellular response to a universal bacteria component is associated with a specific fly behavior, we used the proboscis extension response (PER), which is part of the sensorimotor taste circuit in adult flies[42]. Stimulation of the proboscis by an attractive molecule such as sucrose triggers its immediate and directional extension. When an aversive effect is suspected, the molecule to be tested is combined with a palatable solution such as sucrose. Aversion is measured by the ability of the substance to prevent PER to sucrose. Since our published results[38] demonstrated the ability of PGN to stimulate bitter Gra66a+ neurons, a cell population involved in aversive behaviors[43], we tested whether PGN could suppress the PER response to sucrose. Both Diaminopimelic-type PGN which forms the cell wall of all Gram-negative bacteria and of Bacilli (DAP-PGN; we used here different concentrations of PGN from *Escherichia coli*, *E.c.* 10/100/200 μg/mL), and Lysine-type PGN found in the cell wall of Gram-positive bacteria (Lys-PGN; we used here one concentration of PGN from *Staphylococcus aureus*, *S.a.* 200 μg/mL) were tested[33]. In PER, DAP-PGN was found to be aversive to reference flies (w-) when used at 100 (*E.c.* 100) and 200 μg/mL (*E.c.* 200) but not at lower concentrations (Fig. 1A). These effects were specific to DAP-type PGN as they were not observed with Lys-type PGN (*S.a.* 200, Fig. 1A). Similar aversive effects of DAP-type PGN were obtained with two other commonly used reference *Drosophila* strains, yw and canton-S (Fig. 1B, C). To test the robustness of the

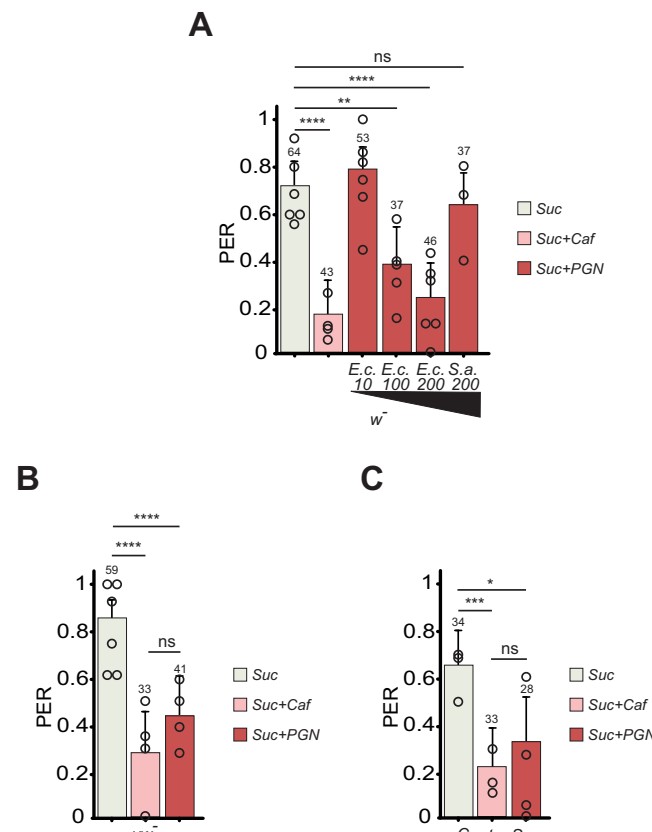

**Fig. 1 | *E. coli* PGN is aversive to the fly.** Dose-dependent PGN inhibition of PER. **A** PER index of *w-* flies to control solutions of sucrose 1 mM and sucrose 1 mM + caffeine 10 mM and to sucrose 1 mM + increasing concentrations of PGN from *E. coli* K12 (*E.c*) and *S. aureus* (*S.a*). The numbers below the x-axis correspond to the PGN final concentrations in μg/mL. **B**, **C** Aversion to PGN is independent of genetic background. PER index of *yw-* (**B**) and CantonS (**C**) flies to control solutions of sucrose 1 mM and sucrose 1 mM + caffeine 10 mM and to sucrose 1 mM + PGN from *E. coli* K12 at 200 μg/mL. The PER index is calculated as the percentage of flies tested that responded with a PER to the stimulation ± 95% confidence interval (CI). The total number of tested flies (n) for one condition is indicated on top of each bar. As the values obtained from one fly are categorical data with a Yes or No value, we used the Fisher exact t-test and the 95% CI to test the statistical significance of a possible difference between a test sample and the related control. At least 3 independent experiments were performed. The results from all the independent experiments were gathered and consequently, we do not show the average response from one experiment representative of the different biological replicates, but an average from all the data generated during the independent experiments in one graph. However, each open circle represents the average PER of one experiment. A PER value of 1 means that 100% of the tested flies extended their proboscis following contact with the mixture, a value of 0,2 means that 20% of the animals extended their proboscis. The number of tested flies (n) is indicated on top of each bar. ns indicates $p > 0.05$, *$p < 0.05$, **$p < 0.01$, ***$p < 0.001$, ****$p < 0.0001$ two-sided Fisher Exact Test. Further details including raw data and exact p-values can be found in the source data file.

effect observed, we quantified it in flies reared on different diets and hence with different metabolic and internal states. Flies reared on both protein-rich (prot + ) or sugar-rich (suc + ) medium displayed an aversion to PGN (Fig S1A and S1B). We used a sucrose concentration that was sufficient to trigger PER in the majority of flies but not too high to prevent aversion to PGN or caffeine. Indeed, as with the well-characterized bitter agent caffeine, the ability of PGN to prevent PER was overcome when PGN was mixed with solutions of higher sucrose concentrations (Fig S1C). As the fly legs are taste organs as well[44], and PER can also be triggered by chemicals coming into contact with tarsi, we tested PGN in this assay. No aversive response to PGN could be

detected in leg-triggered PER, either because the tarsal taste neurons are not involved in the detection of this microbial compound, or because the concentration of sucrose required to trigger the PER on legs overcomes the aversion to PGN (Fig S2A). We, thereafter, focused on PER triggered by direct contact with proboscis. Because taste neurons have been shown to respond to acidic solutions[45,46], we measured the pH of our different PGNs and they were all neutral. This rule out the possibility that the PER response to PGN is due to its acidity (Fig S2B). Taken together, these data demonstrate that DAP-type PGN is perceived as repulsive while Lys-type PGN is not.

### PGN-triggered aversion involves Gr66a+ and ppk23+ neurons

To identify the type of neurons that respond to PGN, PERs were performed in flies in which specific groups of neurons were inactivated by overexpression of the inward rectifier potassium channel Kir2.1 throughout development (larvae, pupae and adult). To test whether the bitter network was involved in PER aversion toward DAP-type PGN, neurons producing the pan-bitter taste receptor Gr66a were inactivated[47] (Fig S3). The ability of DAP-PGN to suppress PER when mixed with sucrose was completely abolished in Gr66a-Gal4/UAS-Kir2.1 flies demonstrating that Gr66a+ cells are necessary to transduce PGN signal (Fig. 2A). Another class of neurons able to mediate aversion in the adults has been described in the taste sensilla[48], they are characterized by the expression of the *ppk23* gene (ppk23+) (Fig S4). We tested their putative implication in mediating PGN aversion using the Kir2.1 overexpression throughout development. Ppk23-Gal4/UAS-Kir2.1 flies no longer perceived PGN as aversive demonstrating that Gr66a+ and ppk23+ neurons are two cell populations necessary to mediate PGN aversion in the tested adults (Fig. 2B). Recent work has shown that flies have 3 neuronal populations expressing either Gr66a and/or ppk23 that display distinct properties related to salt perception[48,49]: (Gr66a+/ppk23+), (Gr66a+/ppk23-) and (ppk23+/Gr66a-)[48,49]. In an attempt to delineate whether one of these play a role in the adult aversion process to PGN, we used a genetic intersectional strategy allowing the visualization or functional manipulation of specific subsets on neurons. When genetically combined, with Gal80 being a Gal4 inhibitor, the ppk23-Gal4 and Gr66a-Gal80 transgenes gave rise to flies in which the Gal4 driver is active only in the ppk23+/Gr66a- neuronal subset[48]. We used this intersectional genetic strategy to visualize and target ppk23+/Gr66a- cells (Fig S5). Inactivation of the ppk23+/Gr66a- population alone throughout development using UAS-Kir2.1 did not alter adult PGN aversion (Fig S6A). This negative result indirectly suggests that the remaining cells, i.e., Gr66a+/ppk23- and/or Gr66a+/ppk23+, may be sufficient to mediate PGN aversion (Fig S6A and S6B).

### Adult ppk23+/Gr66a- cells respond to PGN

While functional data point towards a role of Gr66a+ and ppk23+ neuronal groups in mediating adult aversion to PGN, our attempts to define smaller neuronal subpopulations of Gr66a+ or ppk23+ populations were unsuccessful. To have a more direct readout of the effect of PGN on these cells, we monitored calcium level (using GCaMP) in the sub esophageal zone (SEZ) of the brain of females following proboscis exposure to PGN. This brain area processes gustatory input from gustatory neurons located in the proboscis. Our previous work demonstrated that adult Gr66a+ neurons can respond to PGN[38]. However, as mentioned above, some adult Gr66a+ cells are also ppk23+[48] (Figs. S6A, S6B). In order to avoid the confusing signal from Gr66a+ cells and to assay whether ppk23+ neurons could on their own respond to PGN, we used the intersectional genetic strategy to quantify calcium activity in ppk23+/Gr66a- cells[48]. Results obtained indicate that ppk23+/Gr66a- cells can directly respond to PGN (Fig. 2C–E). Importantly, these cells did not respond to caffeine but did to high salt, a signature expected for ppk23+/Gr66a- cells (Fig. 2E).

### Temporal requirement of gustatory neurons for PGN detection during the fly's lifetime

Functional data demonstrate that Gr66a+ or ppk23+ neurons inactivation throughout development impairs adult response to PGN. In addition, our previous and current calcium-imaging experiments demonstrate that adult labellar Gr66a+ neurons as well as ppk23+/Gr66a- sub populations respond to PGN. The involvement of several neuronal subgroups in PGN detection prompted us to test whether these different neurons are all required at the same time during the fly's life. Indeed, although most studies aimed at dissecting how flies perceive their environment via the taste apparatus are performed at the adult stage, including this one, the bitter neurons exemplified by the Gr66a+ population are present throughout fly development[50,51]. To determine when the neurons necessary for the PGN-induced PER suppression are required, we took advantage of the Gal4/Gal80ts binary system that allowed us to control the inactivation of taste neurons in a spatially and temporally controlled manner (Fig. 3A). Surprisingly, when assessing aversion to PGN, adult flies in which Gr66a+ neurons were functionally inactivated only before the adult stage could no longer perceive PGN as aversive (Fig. 3B). In the parallel experiment, inactivation of Gr66a+ neurons in adult flies only, had no effect. In contrast, perception by the adult flies of quinine and caffeine required Gr66a+ neurons to be functional in the adult but not in larvae (Fig S7A). Thus, concerning the adult aversion to PGN, Gr66a+ neurons are not necessary in the adult, but rather during the larval stage. Since both Gr66a+ and ppk23+ neurons are required to respond to PGN, we tested whether ppk23+ cells would be the ones at play in the adults. The data presented Fig. 3B demonstrated that, unlike Gr66a+ neurons, ppk23+ neurons are functionally required in the adult for the adults PGN-triggered aversion. Furthermore, inactivation of the ppk23+ neurons only during the larval stage did not impair the PGN-triggered aversion in the adults. These results reveal an unexpected link between larval activity of Gr66a+ neurons and the adult capacity to respond to PGN that relies on ppk23+ cells.

Other taste receptors have been described and some are co-expressed with Gr66a+, defining subpopulations of Gr66a+ neurons[43,52]. In an attempt to map more precisely the subset of Gr66a+ neurons responsible for the phenomenon in larvae, we silenced neurons using drivers known to be co-expressed with Gr66a (Fig. 3C). Our results were negative, with none of the Gr66a+ subpopulations silenced impairing adult aversion to PGN. These data do not exclude that the tested cells may be involved and indirectly suggest that the subset of Gr66a+ required for the phenotype may not be defined by co-expression with the drivers we tested (see Discussion). Overall, our data on the temporal inactivation of neuronal groups clarify previous surprising results demonstrating a role for both Gr66a+ cells and ppk23+ cells using inactivation throughout development. Indeed, in regards to PGN-triggered aversion in adults, while Gr66a+ cells are essential during larval life, they are not required at the adult stage. Conversely, ppk23+ neurons are essential during the adult stage and are not required during larval life. The requirement of a neuronal population being ppk23+ and not Gr66a+ (ppk23+/Gr66a-) to trigger adult PGN-triggered aversion is in agreement with our in vivo calcium imaging assays (Fig. 2C–E).

### TRPA1 is functionally required in larval Gr66a+ neurons for a response of adult flies to PGN

Using calcium imaging as a readout, we have shown in a previous study that certain components of the IMD pathway (Fig. 4A), including the upstream transmembrane receptor PGRP-LC, are functionally required for the transduction of DAP-PGN signal in adult Gr66a+ neurons[38]. We therefore, tested whether this well-characterized PGN receptor was also required in ppk23+ cells. PER responses to DAP-PGN were not affected by RNAi-mediated inactivation of PGRP-LC in either ppk23+ or Gr66a+ neurons (Fig. 4B). These results suggesting that DAP-PGN

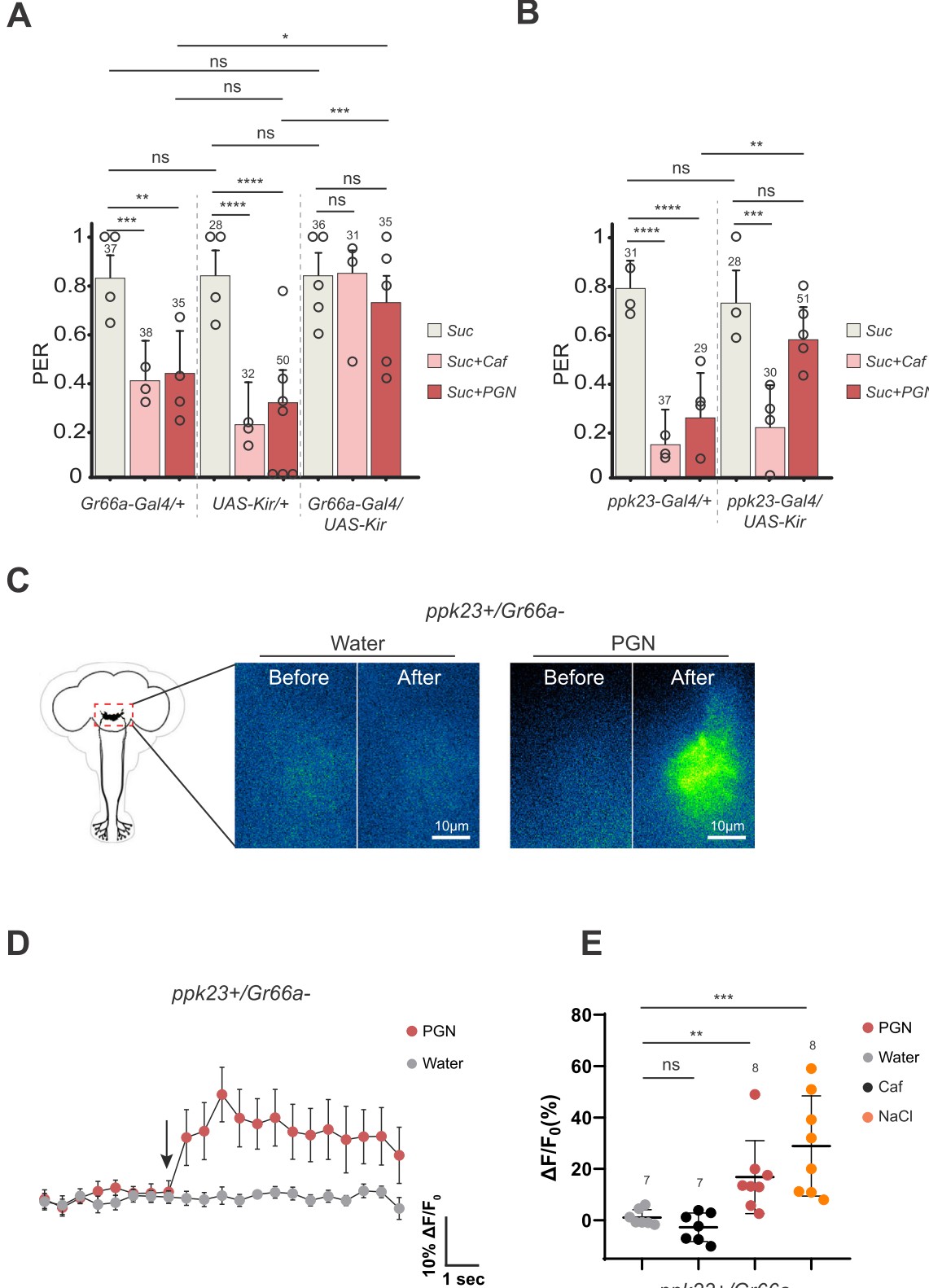

triggered aversion is independent of the IMD pathway were confirmed using RNAi against other components of the pathway (Figs. S8A, S8B).

Since RNAi-mediated downregulation of Myd88, a central component of the Toll pathway did not modify PER response to PGN (Fig. 4C), this other NF-κB pathway is unlikely to be involved in adult PGN-triggered aversion. To clearly rule out an implication of the Toll pathway, more elements of the signaling cascade should be tested.

The Gr66a receptor was also tested for its involvement in PGN-induced PER suppression as it is considered, together with Gr33a, as the main co-receptor for the detection of bitter molecules[53]. While *Gr66a* mutants lost PER aversion for caffeine, they did not for PGN (Fig S8C). In addition to bitter taste receptors, the transient receptor potential TRPA1 channel has been implicated in the detection of bacterial lipopolysaccharide (LPS) by Gr66a+ cells[54,55]. Its RNAi-mediated

**Fig. 2 | Adult fly aversion to PGN requires functionally active Gr66a+ and ppk23+ neurons.** Impairing the activity of Gr66a+ (**A**) or ppk23+ (**B**) neurons via UAS-Kir2.1 abrogates the aversion to PGN. PER index of flies to control solutions of sucrose 1 mM and sucrose 1 mM + caffeine 10 mM and to sucrose 1 mM + PGN from *E. coli* K12 at 200 μg/mL. **C–E** ppk23 + /Gr66a- neurons respond to stimulation with PGN on the labellum. Real-time calcium imaging using the calcium indicator GCaMP6s to reflect the in vivo neuronal activity of ppk23 + /Gr66a- neurons (Gr66a[LexA]; LexAopGal80; ppk23[Gal4]/UAS-GCaMP6s) in adult brains of flies whose proboscis has been stimulated with PGN. The expression of LexAop-Gal80 antagonizes the activity of Gal4, thus preventing the expression of GCaMP6s in Gr66a + / ppk23+ neurons. **C** Representative images showing the GCaMP6s intensity before and after addition of either the control water or the peptidoglycan. **D** Averaged ± SEM time course of the GCaMP6s intensity variations (ΔF/F0 %) for ppk23 + /Gr66a-

neurons. The addition of water (*n* = 7 flies) or peptidoglycan at 200 μg/mL (*n* = 8 flies) at a specific time is indicated by the arrow. **E** Averaged fluorescence intensity of negative peaks ± SEM in response to water (*n* = 7), caffeine 10 mM (*n* = 7), Sodium chloride 250 mM (*n* = 8) or peptidoglycan from *E. coli* K12 at 200 μg/mL (*n* = 8). For (**A**, **B**), PER index is calculated as the percentage of flies tested that responded with a PER to the stimulation ± 95% CI. The number of tested flies (*n*) is indicated on top of each bar. For each condition, at least 3 groups with a minimum of 10 flies per group were used. ns indicates $p > 0.05$, * indicates $p < 0.05$, ** indicates $p < 0.01$, *** indicates $p < 0.001$, **** indicates p < 0.0001 two-sided Fisher Exact Test. In (**E**) ns indicates $p > 0.05$, ** indicates $p < 0.01$, *** indicates $p < 0.001$, non-parametric t-test, two-tailed Mann-Whitney test. Further details including raw data and exact *p*-values can be found in the source data file.

inactivation in Gr66a+ neurons, but not in ppk23+ neurons, abolished the ability of adult flies to respond to PGN (Fig. 4D). These results were refined using stage-dependent inactivation (Fig. 4E) and demonstrate that the expression of TRPA1 channel in larval Gr66a+ neurons is required for the adults that hatch from these larvae to respond to PGN.

### Germ-free flies are not able to trigger PER aversion to PGN

The involvement of multiple temporal and spatial inputs for PGN detection led us to try to identify the nature of the larval-sensed trigger(s) that prepare the adult response to PGN. Since larvae are born and live in contaminated environments, and the larval microbiota has a strong impact on host immunity[56], physiology[57] and behavior[58], we tested its necessity for the adult fly response to bacterially-derived PGN. When animals were reared on antibiotic (germ-free) medium throughout development (larval life and adult life), the number of bacteria per fly, quantified by CFU plating, was strongly reduced (Fig. 5A). Interestingly, while germ-free flies were still able to respond to caffeine their response to PGN was completely abolished (Fig. 5B). Similar results were obtained using different genetic backgrounds demonstrating the universality of the observed effects (Figs. S9A, S9B). The PER suppression in response to PGN was also abolished when flies were made axenic by bleaching the eggs (Figs. S9C, S9D) eliminating possible side effects of antibiotics on fly taste. These results demonstrate that while exposure to bacteria is not mandatory for the adult response to caffeine, animals must be exposed to bacteria for the adults to perceive PGN as aversive. To further confirm the inability of axenic adult flies hatched from germ-free larvae to respond to PGN we used calcium imaging. Whereas calcium levels were increased in ppk23 + /Gr66a- neurons following PGN exposure to the proboscis of conventionally reared flies (Fig. 2C–E), this was no longer the case for germ-free flies (Fig. 5C–E). To ensure that germ-free conditions do not, in a non-specific manner, lower the activity potential of neurons, we quantified the salt response that is dependent upon ppk23+ cells. The salt-related activation of these ppk23 + /Gr66a- neurons monitored by calcium-imaging was also not affected by germ-free conditions, confirming the specific effect of germ-free conditions upon PGN sensitivity (Fig. 5E). These results combined to our temporal genetic inactivation assays suggest a model where larval Gr66a+ neurons have to be exposed to bacteria for the adult ppk23+ neurons to respond to PGN. We verified that activation of Gr66a+ adult neurons by PGN[38] is involved in a process different from the PER by testing germ-free and conventionally reared animals. As shown in Fig. S9E, adult Gr66a+ cells were activated by PGN in both germ-free and conventional conditions. This demonstrate that while adult Gr66a+ cells can respond to PGN regardless of the fly's life experience with bacteria, adult ppk23+ neurons can respond to PGN, only if they hatch from conventionally reared larvae.

### Larval microbiota is a pre-requisite to implementing the adult gustatory response to PGN

Because flies are in contact with environmental bacteria throughout development and later in adulthood, we sought to identify the

temporal window during which the presence of bacteria impacts the adult gustatory system's response to PGN. For that purpose, animals were reared on conventional medium and transferred to antibiotic-containing medium at different periods of their life cycle and for different durations (Fig. 6A). All emerged flies were then tested for their ability to respond to PGN. Whereas flies emerged from conventional larvae reared immediately after hatching on antibiotic medium responded adversely to PGN (Fig. 6B), those from larvae reared on germ-free medium lost this ability, despite adults' exposure to conventional environment (Fig. 6B). To further reduce the permissive time window, larvae were reared on a conventional medium for only a part of the larval period. The efficacy of the different treatments were monitored using CFU tests (Fig S10A). Exposure of larvae to antibiotic food for the 72 first hours of development was sufficient to abolish the aversive response of the adult to PGN (Fig. 6C).

Similarly, while rearing larvae from germ-free eggs on conventional medium for the first 48 h of development was sufficient to restore PGN responsiveness in adults, exposing larvae to conventional medium from 48 h to 96 h only was not (Fig. 6D).

Furthermore, exposure of axenic larval pupae to bacteria (Figs. S10B, S10C), as well as exposure of germ-free larvae and adults to DAP-type PGN-containing media (Fig S11A) was not sufficient for the adult's PGN-induced PER suppression to be re-established. These results indicate that cohabitation of early larvae with bacteria is a prerequisite for the adult taste aversive response to PGN to be established. In addition, these data are consistent with a role for Gr66a+ taste neurons only during larval life for the adult to respond adversely to PGN in a PER assay. Based on our data, a simple model would be that activation of Gr66a+ during larval life, directly or indirectly linked to cohabitation with bacteria, is sufficient to generate adults with the ability to avoid PGN. To test this hypothesis, we ectopically activated the Gr66a+ neurons during larval life only using over-expression of a heat-sensitive TRPA1 isoform. Expression of this construct in neurons and exposition of animals to temperatures above 23 °C trigger a calcium influx that can activate the neuron despite the absence of endogenous stimulation. Thus, constitutive activation of Gr66a+ neurons in germ-free larvae may be sufficient to obtain PGN responsive adults. We performed the test with germ-free larvae and observe no aversion to PGN in the resulting adults (Fig. S11B). Thus, activation of Gr66a+ neurons under our conditions is not sufficient. The type and intensity of activation that we used may not mimic the physiological ones. Alternatively, the system may require multiple inputs, the activation of Gr66a+ larval cells being only one of them.

### Mono-association of larvae with *Lactobacillus brevis* is sufficient to restore the aversive adult response to PGN

Although much simpler than that of mammals, previous work has shown that the *Drosophila* microbiota can host about 20 species mainly belonging to the genus *Acetobacter* or *Lactobacillus*,

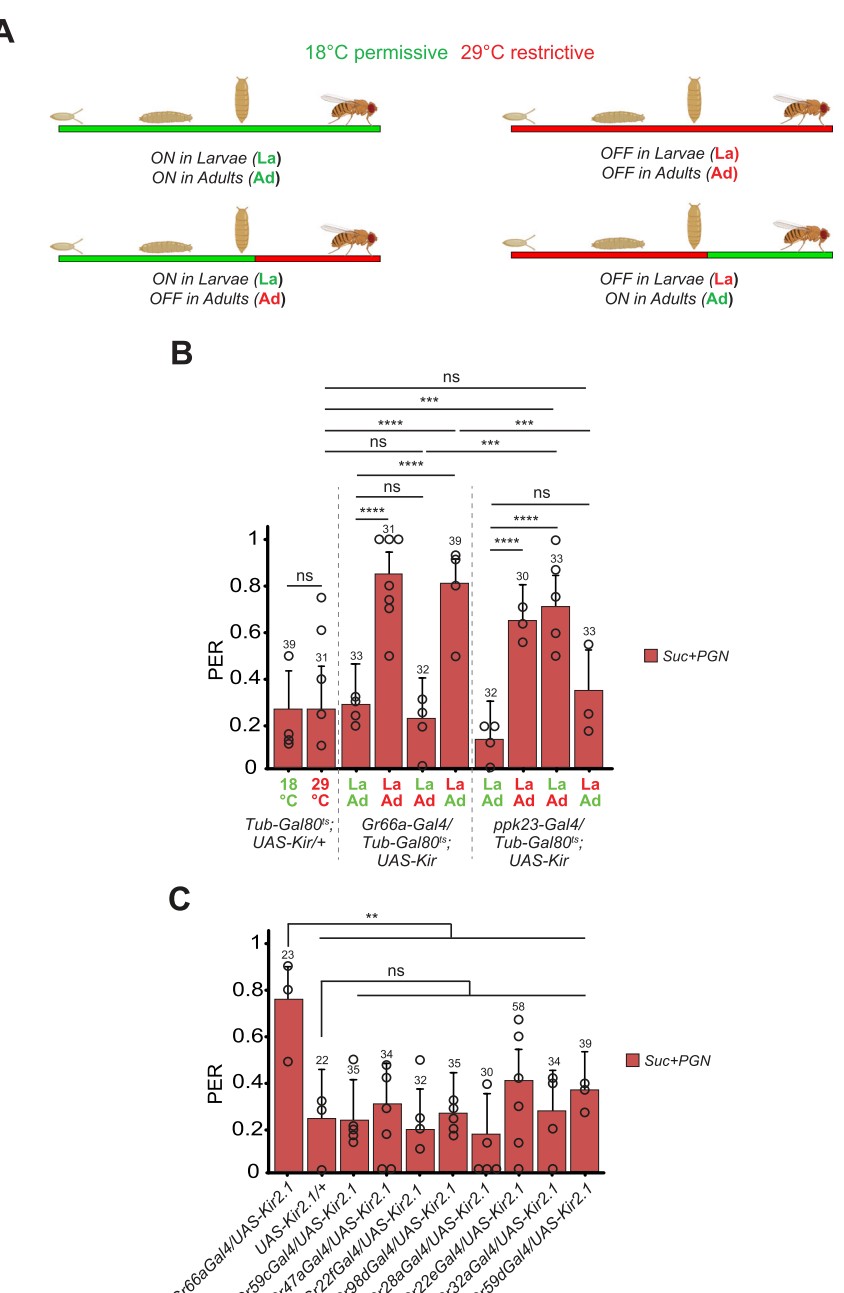

**Fig. 3 | Adult fly aversion to PGN requires functionally active larval Gr66A+ and adult ppk23+ neurons. A** Graphical representation of the life periods during which flies are shifted from 18 °C (green) to 29 °C (red). **B** Gr66a+ neurons are functionally required in the larval stage for PGN-triggered aversion, but are dispensable during adult life. In opposition to ppk23+ neurons which are required in the adult but not during the larval stages. PER index of flies to sucrose 1 mM + PGN from *E. coli* K12 at 200 μg/mL. The ubiquitously expressed Tub-Gal80[ts], that inhibits the activity of Gal4, is temperature sensitive: it's active at 18 °C and inactivated at 29 °C, allowing the expression of UAS-Kir2.1 and the consequent impairment of Gr66a+ or ppk23+

neurons activity. **C** Silencing of smaller subgroup of the Gr66a+ neurons population has no effect on the PGN-induced PER suppression. PER index of flies in which part of Gr66a+ neurons are inactivated via different Gr drivers guiding the expression of Kir2.1, to solutions of sucrose 1 mM + PGN from *E. coli* K12 at 200 μg/mL. For (**B**, **C**), PER index is calculated as the percentage of flies tested that responded with a PER to the stimulation ± 95% CI. The number of tested flies (*n*) is indicated on top of each bar. ns indicates $p > 0.05$, ** indicates $p < 0.01$, *** indicates $p < 0.001$, **** indicates $p < 0.0001$ two-sided Fisher Exact Test. Further details including raw data and exact *p*-values can be found in the source data file.

depending of the fly's diet[59–63]. To test whether a specific bacterial species could mediate larval priming effects, we sequenced the bacteria present in our laboratory fly colony (Fig. 7A). Of the few species identified, *Lactobacillus brevis* (*L. brevis*) was one of the most abundant and caught our attention because it has been shown to affect the behavior of the flies it inhabits[64,65]. Strikingly, PGN-induced PER suppression was restored in axenic adults obtained from larvae mono-associated with *L. brevis* (Fig. 7B, C). When the same experiment was performed with a related species,

*Lactobacillus plantarum* (*L. plantarum*), there was a trend toward PER inhibition, but not as robust as the one we quantified using mono-association with *L. brevis* (Fig. 7C). To test whether *L. brevis* could directly activate Gr66a+ neurons, we monitored calcium levels in Gr66a+ larval neurons exposed to this bacterium. In line with its ability to initiate priming, *L. brevis* was able to induce calcium rise in larval Gr66a+ neurons (Fig. 7D–F). It has been previously reported that the presence of *L. brevis* in adult animals can alter the fly behavior by modulating the production of octopamine,

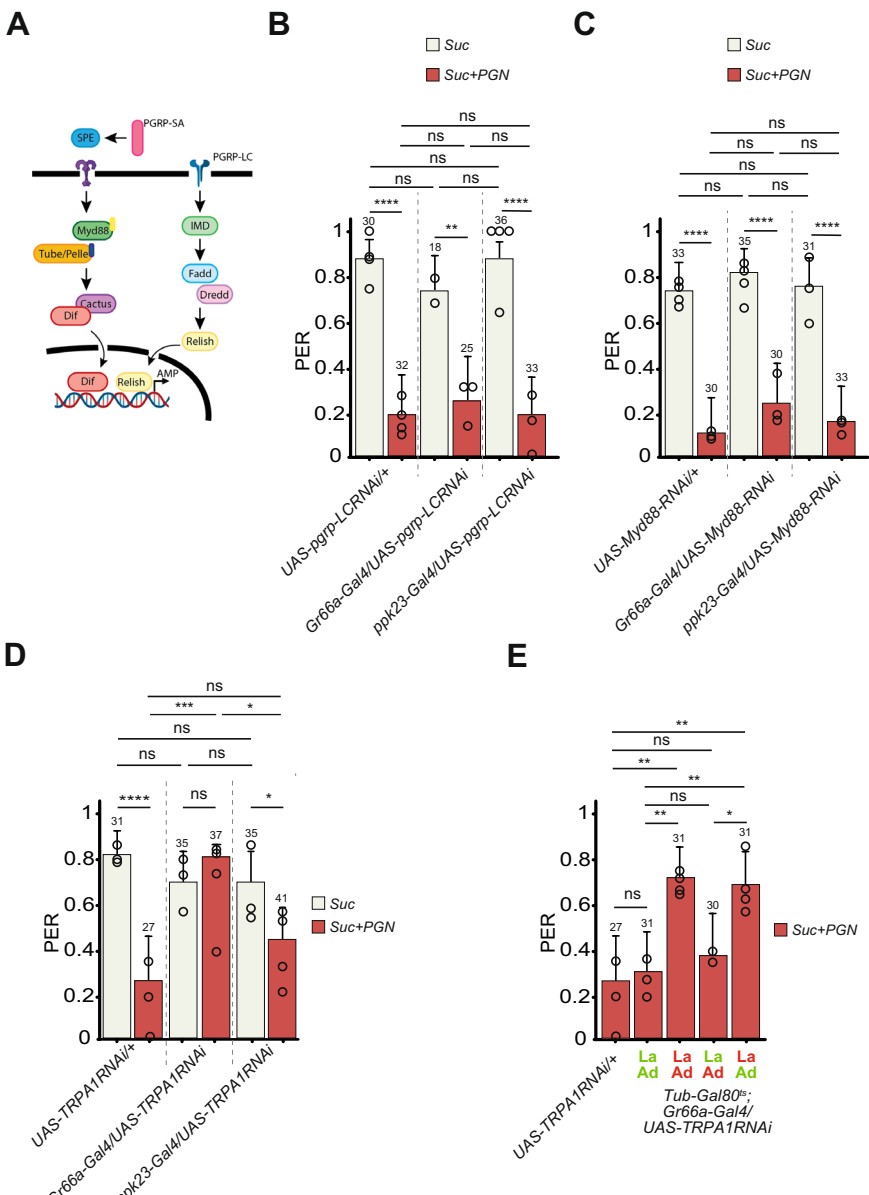

**Fig. 4 | Adult fly aversion to PGN requires TRPA1 expression in larval Gr66a+ neurons. A** Graphical representation of the IMD pathway. **B** RNAi-mediated *PGRP-LC* (UAS-*pgrp-LC* RNAi) inactivation in the Gr66a+ or ppk23+ cells does not affect PGN-triggered aversion. **C** RNAi-mediated *Myd88* (UAS-Myd88 RNAi) inactivation in the Gr66a+ or ppk23+ cells does not affect PGN-triggered aversion. **D** RNAi-mediated TRPA1 (UAS-*TRPA1* RNAi) inactivation in the Gr66a+ cells abrogates PGN-induced aversion, while its inactivation in ppk23+ cells has no effect on the aversion phenotype. **E** The nociceptive TRPA1 channel is required in the larval stages for PGN-induced aversion, while its RNAi dependent inactivation in the adult stage has no effect on PGN-triggered aversion. The ubiquitously expressed Tub-Gal80^ts, that

inhibits the activity of Gal4, is temperature sensitive: it's active at 18 °C and inactivated at 29 °C, allowing the expression of *TRPA1*-RNAi and the consequent inactivation of TRPA1 in Gr66a+ cells. For (**B**–**E**) PER index of flies to control solutions of sucrose 1 mM and to sucrose 1 mM + PGN from *E. coli* K12 at 200 μg/mL. PER index is calculated as the percentage of flies tested that responded with a PER to the stimulation ± 95% CI. The number of tested flies (*n*) is indicated on top of each bar. ns indicates $p > 0.05$, * indicates $p < 0.05$, ** indicates $p < 0.01$, *** indicates $p < 0.001$, **** indicates $p < 0.0001$ two-sided Fisher Exact Test. Further details including raw data and exact *p*-values can be found in the source data file.

a neurotransmitter involved in oviposition, male fighting and locomotor activity[58,65,66]. In our case, exposure of adults to conventional media containing bacteria was not sufficient to restore the PGN PER phenotype (Fig. 6B) and we ruled out the putative involvement of octopamine by demonstrating that the PGN is still perceived as aversive by mutants unable to synthetize octopamine (TβH mutant)[67] (Fig. S11C).

Overall, these results demonstrate that a period of cohabitation between larvae and specific bacterial species, such as *L. brevis*, is mandatory for the emerging adult to perceive PGN as an aversive molecule by ppk23+ neurons.

## Discussion

The data presented here demonstrate that adult flies are competent to perceive and respond to PGN of bacterial origin via ppk23+ neurons (Fig. 3B). However, this behavior is present only in adult flies hatched from larvae carrying functional Gr66a+ neurons and reared in the presence of certain bacterial species (Figs. 3B, 6B and 7C). While the cohabitation of larvae with a bacteria considered as pathobiont like *L. brevis*[68], makes the adults from which they are produced wary of PGN (Fig. 7C), this is less the case for the commensal bacterium *L. plantarum*. This suggests that, depending on the type of bacteria with which the larva comes into contact, it may give rise to adults with a different

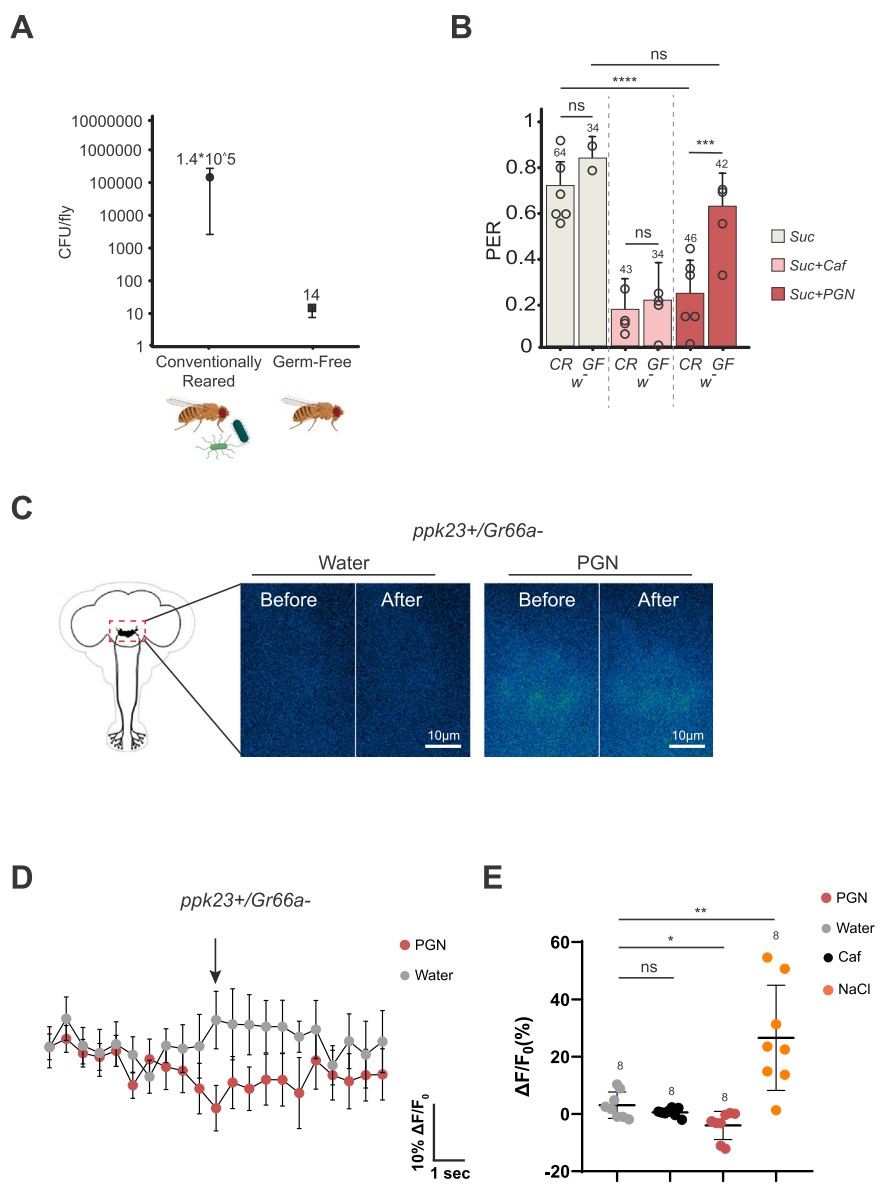

**Fig. 5 | Germ free adult flies do not perceive PGN as bitter. A** CFU/fly comparing conventionally raised flies to those raised on antibiotic media. Antibiotic treatment is effective in reducing the fly bacterial load. **B** Germ-free flies do not show PGN-induced PER suppression. PER index of *w-* Germ-free (GF) and conventionally raised (CR) flies to control solutions of sucrose 1 mM and sucrose 1 mM + caffeine 10 mM and to sucrose 1 mM + PGN from *E. coli* K12 at 200 µg/mL. The data showed for CR flies are the same as in Fig. 1A. For (**B**), PER index is calculated as the percentage of flies tested that responded with a PER to the stimulation ± 95% CI. The number of tested flies (*n*) is indicated on top of each bar. For each condition, at least 3 groups with a minimum of 10 flies per group were used. ns indicates *p* > 0.05, * indicates *p* < 0.05, ** indicates *p* < 0.01, *** indicates *p* < 0.001, **** indicates *p* < 0.0001 two-sided Fisher Exact Test. **C–E** Real-time calcium imaging using the calcium indicator GCaMP6s to reflect the in vivo neuronal activity of Gr66a-/ppk23+ neurons (Gr66a^LexA; LexAopGal80; ppk23^Gal4/UASGCaMP6s) in adult brains of flies whose proboscis has been stimulated with PGN. The expression of LexOp-Gal80 antagonizes the activity of Gal4, thus preventing the expression of GCaMP6S in Gr66a + /ppk23+ neurons. **C** Representative images showing the GCaMP6s intensity before and after addition of either the control water or the peptidoglycan. **D** Averaged ± SEM time course of the GCaMP6s intensity variations (ΔF/F0 %) for Gr66a-/ppk23+ neurons. The addition of water (*n* = 8) or peptidoglycan (*n* = 8) at a specific time is indicated by the arrow. **E** Averaged fluorescence intensity of positive peaks ± SEM of germ-free flies in response to Water (*n* = 8), Caffeine 10 mM (*n* = 8), Sodium chloride 250 mM (*n* = 8) and peptidoglycan from *E. coli* K12 at 200 µg/mL (*n* = 8). In (**E**), ns indicates *p* > 0.05, * indicates *p* < 0.05, ** indicates *p* < 0.01, non-parametric t-test, two-tailed Mann-Whitney test. Further details including raw data and exact *p*-values can be found in the source data file.

perception of the world around them. From an ecological point of view, the presence of pathogenic bacteria in the larval environment would induce a defense mechanism enabling the adult to which they give rise to perceive this environment as potentially dangerous. Such a mechanism should be advantageous. However, it is clear that the adult's behavior is the result of the integration of a multitude of signals emitted by the bacteria and perceived by the fly's gustatory and olfactory systems. Some of these signals, and sometimes the sensory

cells that detect them and the molecular signals they trigger in the host, have already been identified[27,38,54,69,70].

The sensilla of the proboscis are the main taste detectors and act as straws sampling the outside world. Divided into three classes (S, L, I), they each house several cells, including taste neurons whose dendrites are exposed to the outside world via the sensilla opening[71]. While Gr66a + /ppk23- neurons are hosted in I- and one subset of S-sensilla (called Sb), Gr66a + /ppk23+ neurons are found in the other subset of

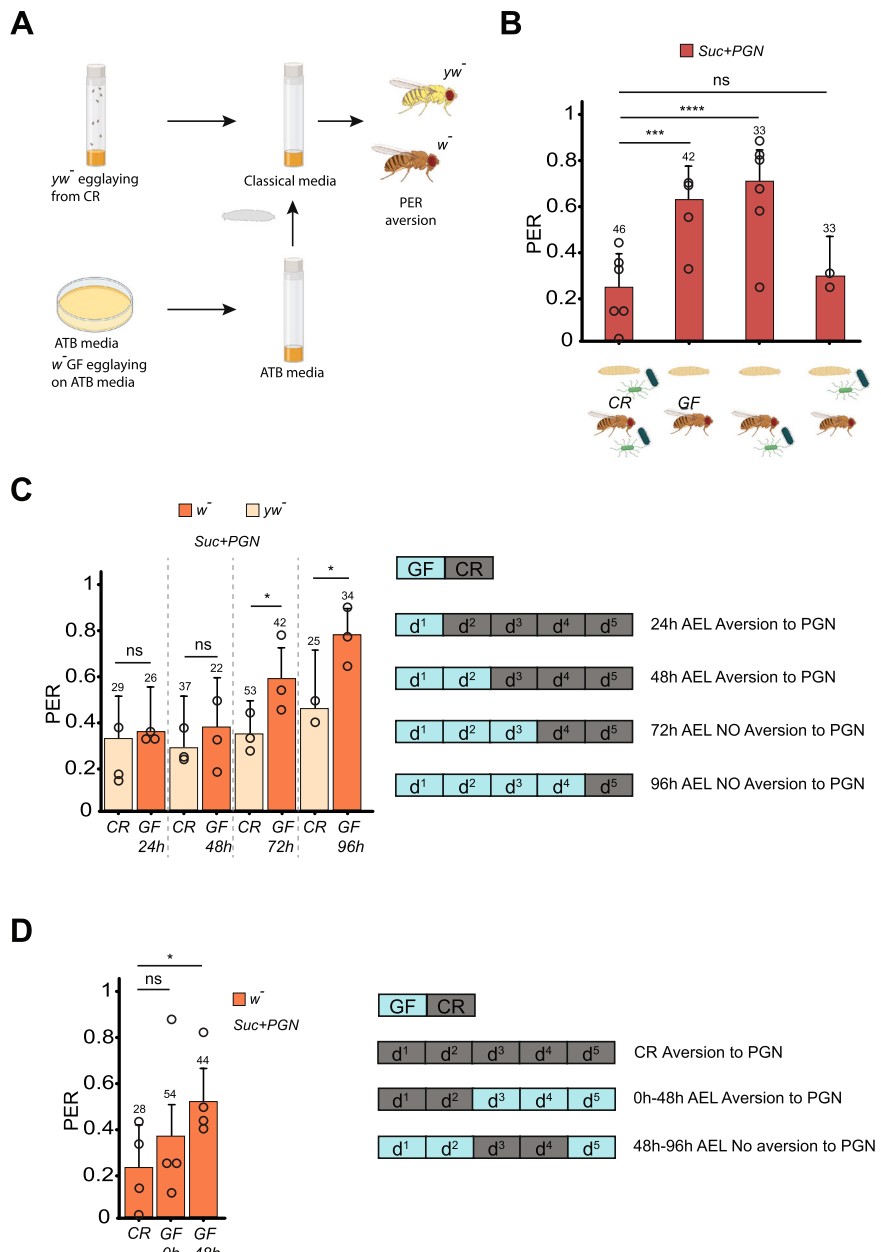

**Fig. 6 | Larval microbiota is required to prime adult gustatory response to PGN.**
**A** Graphical representation of the protocol to contaminate Germ-free larvae at different stage of the larvae development. Two ovipositions of *yw*⁻ flies on standard medium and *w*⁻ flies on antibiotic-enriched medium were started simultaneously. After 4 h the embryos of the *w*⁻ flies are sterilized using bleach and transferred onto antibiotic-enriched media. Upon reaching the desired stage of development the *w*⁻ germ-free larvae are transferred to the same tube in which the *yw*⁻ larvae are growing. Once the adult stage is reached, flies are separated according to body color and their PER response assayed. **B** Co-habitation with bacteria only during the larval stage is sufficient to trigger PGN-induced PER suppression. PER index of *w*⁻ flies, shifted from germ-free to conventional raising conditions (and the reverse) upon pupation, to control solutions of sucrose 1 mM + PGN from *E. coli* K12 at 200 μg/mL. The data showed for conventionally raised (CR) and germ-free (GF) flies are the same as in Figs. 1A and 5B. **C** PER index of *w*⁻ Germ-free animals, shifted to

conventional raising conditions at different stages of their larval development (as in Fig. 6A), to solutions of sucrose 1 mM + PGN from *E. coli* K12 at 200 μg/mL. The graphic represents how many days after the egg laying (AEL) larvae were transferred to *yw*⁻ used media, and their response to stimulation with PGN. **D** PER index of *w*⁻ germ-free animals, transferred for short periods of time of their development under conventional raising conditions, to solutions of sucrose 1 mM + PGN from *E. coli* K12 at 200 μg/mL. The graph represents the timing of the shift from one condition to another. For (**B**–**D**), PER index is calculated as the percentage of flies tested that responded with a PER to the stimulation ± 95% CI. The number of tested flies (*n*) is indicated on top of each bar. For each condition, at least 3 groups with a minimum of 10 flies per group were used. ns indicates $p > 0.05$, * indicates $p < 0.05$, *** indicates $p < 0.001$, **** indicates $p < 0.0001$ two-sided Fisher Exact Test. Further details including raw data and exact *p*-values can be found in the source data file.

S-sensilla (called Sa). ppk23 + /Gr66a- cells are present in few I-sensilla, almost all S- and all L-sensilla[48]. Therefore, some sensilla host both Gr66a + /ppk23+ and ppk23 + /Gr66a- cells.

Since our data show that adult Gr66a+ neurons are dispensable and adult ppk23+ cells are essential for PGN-mediated PER, we suspect

that ppk23 + /Gr66a- cells are important in the process. However, although the ppk23 + /Gr66a- subset may respond to PGN (Fig. 2C–E), flies in which ppk23 + /Gr66a- cells are silenced remain capable of reacting to the PGN via a reduced PER (Fig. S6A). One possibility would

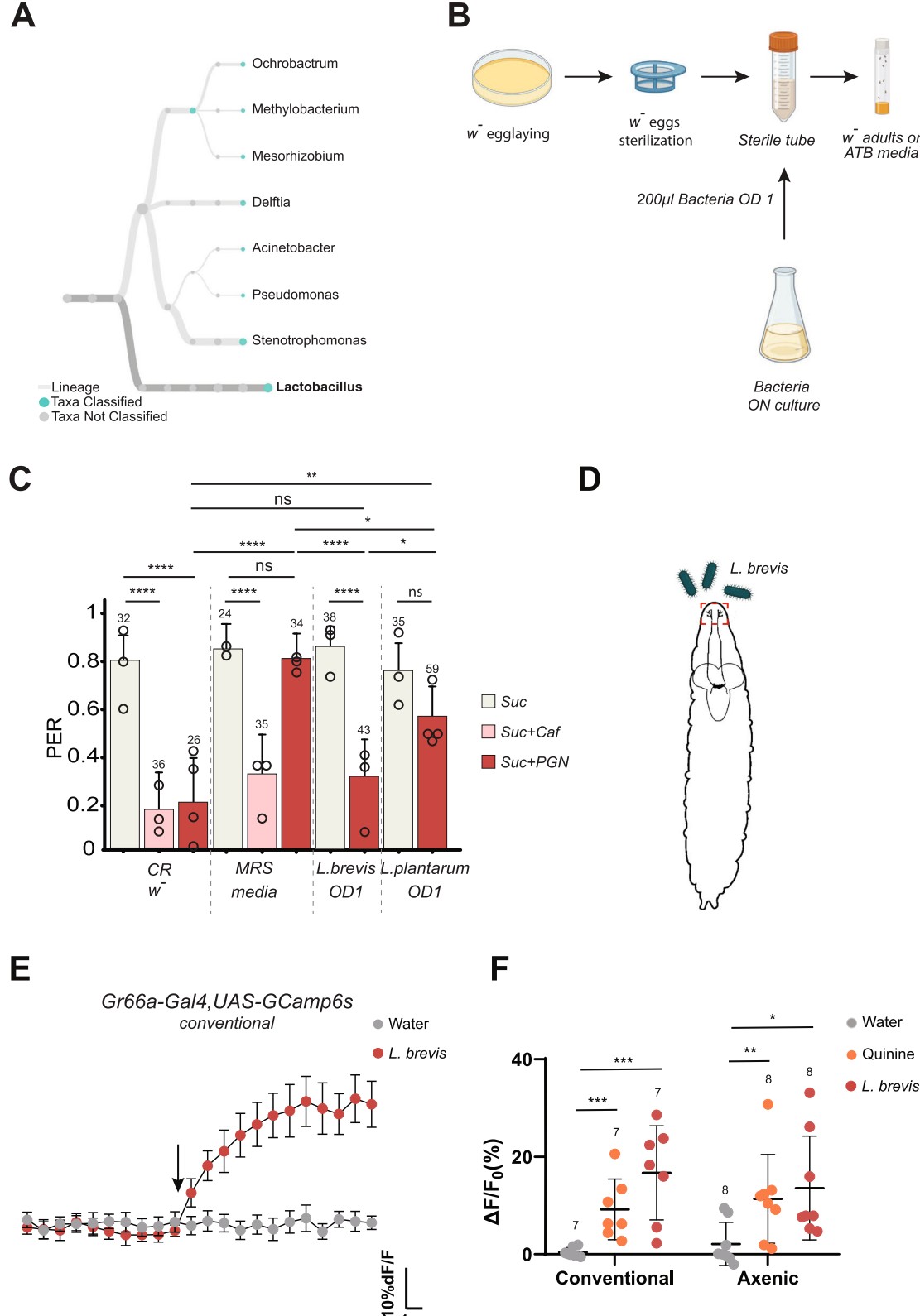

be that ppk23 + /Gr66a+ and ppk23 + /Gr66a- neurons could perform the task in adult (Fig. S6).

The description of new receptors expressed on specific subsets of ppk23 + /Gr66a- neurons such as Ir7c should help to understand the complexity of this taste language[72]. A recent study related to salt detection illustrates this complexity. While high concentrations of monovalent salt triggers a response in both Gr66a+ and ppk23+

neurons, Ir7c is essential for this response mainly in ppk23+ neurons[72]. As with adults, the minimal subset of cells required in larvae to prime the adult response to PGN remains partially elusive. Larval Gr66a+ neurons can be divided into Gr32a+ and Gr32a- subpopulations[52]. Gr32a/Kir2.1 animals respond to PGN like controls, confirming that adult Gr66a+ neurons, which are all Gr32a+ in flies[73], are dispensable for PGN aversion to PER (Fig. 3C). This also suggests the involvement of

**Fig. 7 | Larval colonization by Lactobacillus brevis is sufficient to prime adult gustatory response to PGN. A** Lactobacilli are among the most abundant genera in our conventional raising conditions. Graphical representation of the bacterial genera present in our conventional raising conditions according to 16 S sequencing. **B** Graphical representation of the protocol for the mono-association of germ-free larvae with a given bacterial species. After oviposition the embryos are sterilized with bleach and transferred under sterile conditions into a fresh sterile tube then 200 μL of bacterial culture at OD 1 is added. Adults emerging from pupae are then transferred on antibiotic-enriched media. **C** Mono association of Germ-free larvae with *L. brevis* is sufficient to obtain adults responsive to PGN. PER index of *w-* flies mono-associated with *L. brevis*, *L. plantarum* or exposed to their culture media (MRS) to sucrose 1 mM and sucrose 1 mM + caffeine 10 mM and to sucrose 1 mM + PGN from *E. coli* K12 at 200 μg/mL. **D** Representation of a late L2 larvae with schematic of the brain and projections from taste neurons at the periphery to the CNS. In red is the area chosen to image the GCaMP6s signal following exposure to

water, quinine or *L. brevis*. **E** Averaged ± SEM time course of the GCaMP6s intensity variations (ΔF/F0 %) for Gr66a+ neurons in conventionally raised larvae. The addition of water ($n = 7$), or *L. brevis* (OD1) ($n = 7$) at a specific time is indicated by the arrow. **F** Averaged fluorescence intensity of positive peaks ± SEM for Gr66a+ neurons following treatments in conventionally raised (water $n = 7$, Quinine (3 mM) $n = 7$ or *L. brevis* (OD1) $n = 7$) or axenic larvae (water $n = 8$, Quinine (3 mM) $n = 8$) or *L. brevis* (OD1) $n = 8$). For (**C**) PER index is calculated as the percentage of flies tested that responded with a PER to the stimulation ± 95% CI. The number of tested flies (*n*) is indicated on top of each bar. For each condition, at least 3 groups with a minimum of 10 flies per group were used. In (**C**), ns indicates $p > 0.05$, * indicates $p < 0.05$, ** indicates $p < 0.01$, **** indicates $p < 0.0001$ two-sided Fisher Exact Test. In (**F**), * indicates $p < 0.05$, ** indicates $p < 0.01$, *** indicates $p < 0.001$, non-parametric t-test, two-tailed Mann-Whitney test. Further details including raw data and exact *p*-values can be found in the source data file.

the larval Gr66a + /Gr32a- subpopulation. Silencing other smaller neuronal subgroups using Gr Gal4 drivers[52], had no impact on PGN aversion in the adult (Fig. 3C). Although our data show that ppk23+ cells are not required during the larval period, the existence of larval Gr66a + /ppk23+ neurons has not been clearly established. All these suggestions are based on experiments that failed to impair the phenotype and are consequently indirect. Further genetic dissection that may provide direct evidences will be necessary to delineate the neuronal sub-populations.

The demonstration that adult ppk23+ neurons respond to PGN prompted us to identify the upstream receptor(s). Despite the expected involvement of the classical immune PGN sensor PGRP-LC, its functional inactivation has no impact on aversion towards the microbial compound. Even elimination of central elements of the signaling pathway, such as Fadd, were without consequences. The lack of involvement of the IMD/NF-kB pathways in the larval priming phase was also indirectly suggested when we targeted Gr66a+ neurons (Fig. 4B, 4C, S8A, and S8B). Our data rather identify TRPA1 as a putative receptor necessary on larval Gr66a+ neurons to sense a ligand related to microbial activity (Fig. 4D, E). As for the adult response, the yet to be identified receptor is expected to be expressed on the dendrites of adult ppk23+ neurons, exposed to the sensilla-bathing medium and hence to PGN coming from exogenous bacteria. Unfortunately, the physiology of ppk23+ neurons is mostly understood via their role in pheromone perception[74] and of high salt avoidance[48,72]. Few surface receptors have been described and include Gr66a, ppk23 and Ir7c. We have shown that Gr66a is not required (Fig. S8C). An alternative approach to identify the receptor could be to decipher the signaling pathway required for PGN-mediated PER in ppk23+ cells. For example, if Gαq is involved, GPCRs would appear as good candidates[75].

Our results highlight how the genetic characteristics of the larva and the environment in which they live can impact the development of a sensory response in the adult to whom they will give birth. For the priming to take place larvae must co-live with bacteria, while their presence during adulthood is dispensable (Figs. 5B and 6B). Interestingly, it has recently been reported that the commensal microbiome must interact with the diet during a critical larval developmental period to shape aggressive behaviors in adult males[58]. In many studies associating behavioral changes to septic conditions, including the former mentioned, the neurotransmitter octopamine (OA) plays a central role. This does not seem to be the case here since *TβH* mutant flies, which do not produce OA, respond well to PGN (Fig. S11C). Recolonization experiment of axenic larvae with single bacteria species highlight another difference between these two studies. While PGN sensing can be robustly primed by *L. brevis* but less by *L. plantarum*, *Acetobacter*, *Lactobacilli*, and *Enterococci*, are equally competent to restore aggressive behaviors in germ free males, suggesting that there are common genetic determinants in these bacteria species that promote adult aggression. Further work will be required to

identify these genetics determinants or bacterial compounds mediating the effects.

Our data point to changes initiated by the interaction between TRPA1 receptor expressed on larval Gr66a+ neurons and bacteria (at least *L. brevis*) present during the first 2-days of the larval life. An important question to address is how such an interaction is taking place. Among many possibilities, one can propose, i) modifications of the media by the bacteria, ii) changes in the gut physiology upon bacteria detection, iii) impact of microbial activity on the host iv) direct effects of some bacterial components. In the latter category, one candidate is the PGN itself which plays key role in the interactions between flies and bacteria. However, although PGN is universally present on all bacteria, only some of them are able to mediate priming as mono association with *L. plantarum* was not sufficient. Furthermore, our unsuccessful attempts to rescue adult response by incubating larvae with purified DAP-type PGN do not support this hypothesis (Fig. S11A). Finally, while the PGN we test in adult is DAP-type, the PGN of *L. brevis* is Lys-type. Intriguingly, while the strain of *L. plantarum* that we used, considered as a symbiont, was not able to fully recapitulate the effect of a septic media, *L. brevis* that is a pathobiont with a distinctive molecular signature involving uracil was sufficient to make competent adults[68,76]. It is interesting to note that Soldano et al. have already implicated the TRPA1 channel in the detection of bacterial LPS in Gr66a + cells[54]. However, as *lactobacilli* are Gram-positive bacteria and do not have a LPS layer, this cell wall component is unlikely to be the stimulus for Gr66a+ larval neurons.

Our data suggest that some pieces of information acquired during the larval life experience are transmitted to the adult. Interestingly this effect shows some degree of specificity. Indeed, while the ppk23+ cell-dependent aversion to PGN is lost, adult ppk23+ neurons from axenic larvae are not lost or anergic as they remain responsive to salt (Fig. 5E). In addition, the activation of neurons directly exposed to the environment and the lack of this activity when the larvae were reared in germ-free conditions suggests that the changes occur cell-autonomously within the ppk23+ neurons and not indirectly via other neurons. However, the situation might be more complex as the in vivo imaging is performed in the brain where multiple axons converge making the resolution at the neuron-scale difficult (Fig. 2C). Indeed, a subset of ppk23 + /Gr66a- cells at the periphery might be dedicated to PGN-sensing and not especially to salt, and fully incapacitated by axenic rearing while the others only respond to salt, a capacity unaffected by microbial exposure.

This study raises the question of the mechanisms by which information acquired by larvae in contact with bacteria are specifically transmitted to adults. Since larval Gr66a+ neurons are necessary for this process, it is conceivable that they persist in the adult. However, like most taste neurons that are either lost or transformed during metamorphosis, larval Gr66a+ neurons do not persist into adulthood, except for a subset that becomes adult pharyngeal taste cells[77–80].

However, since PER is performed by sensory touch without ingestion and adult Gr66a+ cells are dispensable for the phenotype, this is unlikely to be the case. Another possibility would be that some larval Gr66a+ cells become adult ppk23+ neurons. The activation state of the larval GR66a+ neurons will then have a direct impact on that of the adult ppk23+ neurons.

A long-term memory or a memory maintained throughout metamorphosis or pre-imaginal conditioning could be considered. Definitive proofs of such mechanisms are still lacking since chemicals used to condition and train the larvae could remain in trace form in the media or on the resulting pupae and, in term, influence the newborn flies[81,82]. These caveats are not valid here since the spatial and temporal genetic inactivation of neurons and the silencing of TRPA1 demonstrate the role of these cells specifically and only during larval life (Figs. 3B, 4D, and E). In addition, the effect of time-controlled bacterial exposure and the lack of response to PGN in adults exposed for five days or from pupae to septic media support the idea that larvae must be reared in non-axenic conditions to produce PGN responsive adults. Bacteria-larvae interactions taking place during the priming period could activate some kind of memory. Such hypothesis is not supported by a recent study that tracked the fate of neurons involved in memory finding no anatomical substrate for a memory trace persisting from larva to adults[80]. Conditions altering memory consolidation in general will be tested in the future[83].

Epigenetic modifications such as histone mark alterations could be affected by the presence of bacteria in the larval gut. Previous reports indicate that defects associated with histone demethylase kdm5 inactivation can be partially rescued by modulating the gut microbiota either by supplementation with *L. plantarum* or by antibiotic treatment[84]. If epigenetic modifications are involved, they will have to be limited since the changes we detected in adult taste are not generic for all tastants (Fig. 5B). However, such a possibility should not be ruled out as it has already been shown that animal exposure to microbe can influence behavior and vertical transmission of the phenotype via epigenetic modifications[85,86].

Using calcium imaging, we previously demonstrated that PGN exposure impacts neuronal activities in an IMD/NF-κB pathway dependent manner[38]. Both brain octopaminergic neurons and adult Gr66a+ taste neurons present IMD pathway dependent calcium modulation following PGN exposure. In the current report we demonstrate that Gr66a+ cells are dispensable for the behavior in the adult stage (Fig. 3B). Different elements support the fact that the proboscis Gr66a+ cells activation we previously described is linked to a phenotype yet to be fully uncovered. First, when the Gr66a+ are silenced specifically during the adult stage, while the animals no longer respond to caffeine, we still observe the PGN-mediated PER (Figs. 3B and S7A). Second, when the animals no longer react to the PGN for they have been raised in axenic conditions, the Gr66a+ neurons still respond to the presence of PGN (Fig. S9E). Finally, while we previously reported that the IMD pathway was essential for the increased calcium concentration in Gr66a+ neurons following PGN exposure[38], this signaling cascade is not required for the PGN-mediated PER (Figs. 4B, C, S8A, and S8B). In agreement, we've identified that another set of neurons present in the proboscis is necessary and sufficient for the PGN-mediated PER, the ppk23+ neurons. Further work will be needed to understand how flies integrate these responses to PGN and to other bacterial components such as lipopolysaccharide to respond adequately to the presence of bacteria in their immediate surrounding.

## Methods

### Fly husbandry
Flies were grown at 25 °C on a yeast/cornmeal medium in 12 h/12 h light/dark cycle-controlled incubators. For 1 L of food, 8.2 g of agar (VWR, cat. #20768.361), 80 g of cornmeal flour (Westhove, Farigel maize H1), and 80 g of yeast extract (VWR, cat. #24979.413) were cooked for 10 min in boiling water. 5.2 g of Methylparaben sodium salt (MERCK, cat. #106756) and 4 mL of 99% propionic acid (CARLOERBA, cat. #409553) were added when the food had cooled down. Suc + food recipe. For 1 L of food, 11 g of agar (VWR, cat. #20768.361), 80 g of cornmeal flour (Westhove, Farigel maize H1), 20 g of yeast extract (VWR, cat. #24979.413) and 30 g of Sucrose were cooked for 10 min in boiling water. 2.5 g of Moldex and 5 mL of 99% propionic acid (CARLOERBA, cat. #409553) were added when the food had cooled down.

### PER assay
All flies used for the test were females between 5 and 7 days old. Unless experimental conditions require it, the flies are kept and staged at 25 °C to avoid any temperature changes once they are put into starvation. The day before, the tested flies are starved in an empty tube with water-soaked plug for 24 h at 25 °C.

Eighteen flies are tested in one assay, 6 flies are mounted on one slide and in pairs under each coverslip. To prepare the slide, three pieces of double-sided tape are regularly spaced on a slide. Two spacers are created on the sides of each piece of tape by shaping two thin cylinders of UHT paste. To avoid the use of carbon dioxide flies are anesthetized on ice. Under the microscope, two flies are stuck on their backs, side by side, on same piece of tape so that their wings adhere to the tape. A coverslip is then placed on top of the two flies and pressed onto the UHT paste, blocking their front legs and immobilizing them.

Once all slides are prepared, they are transferred to a humid chamber and kept at 25 °C for 1.5 h to allow the flies to recover before the assay.

Flies are tested in pairs, the test is carried out until completion on a pair of flies (under the same coverslip), and then move on to the next pair.

Before the test, water is given to each pair of flies to ensure that the flies are not thirsty and do not respond with a PER to the water in which the solutions are prepared. Stimulation with the test solution is always preceded and followed by a control stimulation with a sweet solution, to assess the fly's condition and its suitability for the test. During the test small strips of filter paper are soaked in the test solution and used to contact the fly's labellum (three consecutive times per control and test phase). Contact with the fly's proboscis should be as gentle as possible. Ideally the head should not move. A stronger touch may prevent the fly from responding to subsequent stimulation. Based on the protocol needed the test is done following the sequence and the timing in the table below (Table 1).

All solutions to be tested are prepared the test day and stored at room temperature. In the aversion protocol, the control stimulation is performed with 1 mM sucrose (D(+)-sucrose ≥ 99.5%, p.a. Carl Roth GmbH + Co. KG). This concentration is sufficient to elicit a PER but is not so high as to influence the response to the subsequent test stimulation. Caffeine is from Sigma #C0750-100G and Quinine is from Sigma #Q1125-5G. After each control or test stimulation, a water-soaked strip is used to tap the proboscis and clean it.

The response of the fly to each stimulation is recorded and averaged. Flies that respond positively (PER) to at least one of the control stimulations are considered for further analysis, the others are

## Table 1 | Aversion protocol

| Water To satiety | Wait 1 min | CONTROL Sucrose 1 mM Repeated 3 times | Wait 1 min | Water one time | Wait 1 min | TEST Repeated 3 times | Wait 1 min | Water one time | Wait 1 min | CONTROL Sucrose 1 mM Repeated 3 times |
|---|---|---|---|---|---|---|---|---|---|---|

discarded. The PER index is calculated as the percentage of flies tested that responded with a PER to the TEST stimulation and represented as ± 95% CI. In case of stage dependent experiments, flies are shifted from one condition to another upon hatching.

### In vivo calcium imaging

In vivo adult calcium imaging experiments were performed on 5–7 day-old starved mated females. Animals were raised on conventional media with males at 25 °C. Flies were starved for 20–24 h in a tube containing a filter paper soaked in water prior to experiments. Flies of the appropriate genotype were anesthetized on ice for 1 h. Female flies were suspended by the neck on a plexiglass block (2 × 2 x 2.5 cm), with the proboscis facing the center of the block. Flies were immobilized using an insect pin (0.1 mm diameter) placed on the neck. The ends of the pin were fixed on the block with beeswax (Deiberit 502, Siladent, 1345 209212). The head was then glued on the block with a drop of rosin (Gum rosin, Sigma-Aldrich 1346 -60895-, dissolved in ethanol at 70%) to avoid any movements. The anterior part of the head was thus oriented towards the objective of the microscope. Flies were placed in a humidified box for 1 h to allow the rosin to harden without damaging the living tissues. A plastic coverslip with a hole corresponding to the width of the space between the two eyes was placed on top of the head and fixed on the block with beeswax. The plastic coverslip was sealed on the cuticle with two-component silicon (Kwik-Sil, World Precision Instruments) leaving the proboscis exposed to the air. Ringer's saline (130 mM NaCl, 5 mM KCl, 2 mM MgCl$_2$, 2 mM CaCl2, 36 mM saccharose, 5 mM HEPES, pH 7.3) was placed on the head. The antenna area, air sacs, and the fat body were removed. The esophagus was cut without damaging the brain and taste nerves to allow visual access to the anterior ventral part of the sub-esophageal zone (SEZ). The exposed brain was rinsed twice with Ringer's saline. GCaMP6s fluorescence was viewed with a Leica DM600B microscope under a 40x water objective. Stimulation was performed manually using a pipette with gel loading tip by applying 140 μL of tastant solution diluted in water on the proboscis. The gustatory stimulation was continuous as the tastant is in the drop contacting the proboscis. The recording started before the addition of the tastant and the calcium response could be observed immediately following the contact with the sensilla. For each condition, $n = 7–9$.

In vivo larval calcium imaging experiments were performed on late second instar larvae. Larvae were immobilized in a small drop of distilled water placed between two plastic coverslips (22 mm × 22 mm, Agar Scientific). The two coverslips were held together to prevent movement of the anterior part of the larva using an alligator clip attached to a support. Stimulation was performed manually using a pipette with gel loading tip by applying water, *L. brevis* solution or quinine (3 mM, Sigma #Q1250-10G) diluted in water through a hole made in the upper coverslip allowing the solutions to come into contact with the larval Terminal Organ (TO). For *L. brevis* stimulation, bacteria were grown in MRS medium overnight at room temperature, spined down for 15 min at 2500 x g and the pellet was suspended in water to obtain a final optical density (OD) 600 nm of 1. Gr66a neurons were recorded in the TO of heterozygous larvae (Gr66a-Gal4/UAS-GCaMP6s) with a 10x dry objective.

For both larval and adult calcium imaging experiments, GCaMP6s was excited using a Lumencor diode light source at 482 nm ± 25. Emitted light was collected through a 505–530 nm band-pass filter. Images were collected every 500 ms using a Hamamatsu/HPF-ORCA Flash 4.0 camera and processed using Leica MM AF 2.2.9. Each experiment consisted of a recording of 70–100 images before stimulation and 160 images after stimulation. Data were analyzed as previously described[38] by using FIJI (https://fiji.sc/). For larvae fluorescence quantifications, a background fluorescence variation was calculated and subtracted to the fluorescence variation signal.

### Imaging

To image larvae sensory organs and body full larvae were killed in ethanol 70%, rinsed in PBS and mounted whole on slides using Vectashield fluorescent mounting medium. Brain, proboscises, legs, gut and ovaries were dissected in PBS, rinsed with PBS and directly mounted on slides using Vectashield fluorescent mounting medium. The tissues were visualized directly after. Images of brains, larvae, legs and proboscis were captured with LSM 780 Zeiss confocal microscope (20x air objective was used). Images of carcass, gut and ovaries were captured using Leica M205 FA fluorescence stereo microscope (0.5x plan objective was used). Images were processed using Adobe Photoshop.

### Bacterial strains and maintenance

For this study, the following bacterial strains were used: *L. brevis* (F. Leulier's Lab) and *L. plantarum* (A. Gallet's lab). All the strains were grown in 49 mL MRS liquid media (MRS Broth Fluka analytical 69966) static at 37 °C in sealed 50 mL tubes for anaerobic conditions. To concentrate bacteria and reach the requested OD, 250 mL overnight cultures were centrifuged 15 min at 2250 rcf, OD at 600 nm was measured and bacteria were diluted in MRS medium to the desired concentration.

### Flies bacteria load

The flies are anesthetized on ice, ten females for each experimental condition are pulled in 1.5 mL microcentrifuge tubes. Under sterile conditions, 600 μL of Luria-Bertani liquid media (LB) is added to each tube and a sterile pestle is used to homogenize the flies. The fly homogenate is then serially diluted in LB medium and plated in triplicate on LB plates. Plates are kept overnight at 37 °C to facilitate bacterial growth. After two or three days of growth, the average number of colonies per dilution is calculated and the bacterial count of the fly is determined as follows:

CFU/mL = (1000 μL/volume plated) *average n of colonies

CFU/mL at the origin = CFU/mL*dilution factor

CFU/fly = (CFU/mL at the origin * volume in which you homogenized flies)/n of flies homogenized.

### Embryo sterilization

Three petri dishes are filled with 2.6% bleach, 70% ethanol (Ethanol 96° RPE Carlo Erba Ref 414638) and autoclaved purified distilled water respectively. The embryos are collected by filling the plate in which oviposition occurred with PBS and using a small brush to gently detach them from the flies' food. A 40 μm cell strainer is used to collect the embryos. The cell strainer with the embryos is then dipped into: bleach 2.6% for 5 min, ethanol 70% for 1 min, purified water for 1 min, ethanol 70% for 1 min, purified water for 1 min. The brush used to collect the embryos is sterilized in 2.6% bleach for ten min, rinsed and then used to transfer the sterile embryos onto the desired media.

### Larvae contamination

An oviposition of *w-* germ free flies is set up on an apple agar plate at 25 °C for 4 h. Simultaneously an oviposition of *yw-* flies is set up in a regular fly tube. After 4 h the *w-* embryos are sterilized (see protocol above) and transferred on antibiotic enriched media. Once they reach the desired developmental stage *w-* larvae are filtered out of the antibiotic enriched media and rinsed off with PBS. They are then transferred in the tube in which *yw-* larvae are growing. In this way *w-* larvae are exposed to the same bacteria as *yw-* flies but starting at a defined developmental stage. Once the adults emerge from the pupae, they can be transferred to germ-free tubes and they are distinguished on the basis of body color. This way, while *yw-* animals were exposed to the bacteria throughout their lives, *w-* animals are exposed in a controlled manner depending on the moment of transfer.

## Pupae contamination

*yw-* flies are raised in a regular tube, after a few days the flies are removed and the tube is filled with Luria Bertani liquid media (LB) and incubated at 37 °C. The day after the OD is measured and the culture is diluted to OD1. 200 μL of bacterial culture are added on filter paper on top of the fly media. Germ-free fly pupae attached to the cotton plug are soaked in the bacterial suspension and this plug is used to close the previously contaminated tube. This ensures that flies are exposed to the bacteria as soon as they leave the pupal case.

## Exposure of adult flies to PGN

As soon as they emerge from the pupal case, the germ-free adult flies are transferred to a test tube containing filter paper soaked in a Sucrose1mM + PGN200μg/mL solution and placed on top of the fly medium. Flies are then flipped into tubes prepared in the same way, every day for five days and then used for the PER test.

## Exposure of larvae to PGN

An oviposition of *w-* germ free flies is set up on antibiotic culture medium, in parallel with another oviposition of *yw-* flies set up on antibiotic culture medium. Once the fly media has been softened by the *yw-* larvae 500 μl of PGN 200 μg/mL solution are added on top of it together with 48 h old *w-* germ- free larvae. Once emerged from the pupal case adult flies are sorted based on body color and staged for the PER assay on antibiotic enriched fly food.

## Statistics and data representation

**GraphPad Prism 8 software was used for statistical analyses.** For in vivo calcium imaging, the D'Agostino–Pearson test to assay whether the values are distributed normally was applied. As not all the data sets were considered normal, non-parametric statistical analysis such as non-parametric unpaired Mann–Whitney two-tailed tests was used for all the data presented. Some images have been generated using Bio Render (https://www.biorender.com/).

For PER datasets. As the values obtained from one fly are categorical data with a *Yes* or *No* value, we used the Fisher exact t-test and the 95% confidence interval to test the statistical significance of a possible difference between a test sample and the related control.

For PER assays, at least 3 independent experiments were performed. The results from all the experiments were gathered and the total amount of flies tested is indicated in the graph. In addition, we do not show the average response from one experiment representative of the different biological replicates, but an average from all the data generated during the independent experiments in one graph. However, each open circle represents the average PER of 1 experiment.

## Microbiota sequencing

Five replicates of 20 larvae were considered for 16 S metagenomic analysis. DNAs were extracted using DNeasy PowerSoil kit (Qiagen) by grinding the larvae in the C1 buffer of this kit with a Precellys (VWR).

The Oxford Nanopore Technologies 16 S Barcoding Kit (SQK16S-024) was used to amplify and sequence the full-length 16S ribosomal RNA. Starting with 20–25 ng of extracted DNA, 45 cycles of PCR amplification were performed using New England Biolabs LongAmp Hot Start Taq 2X Master Mix and ONT 16 S barcoded primers (27 F and 1492 R). After purification, the amplicons were quantified and qualified to be mixed equimolarly. Sequencing was performed on Mk1C (MinKNOW 21.10.8) using an R9 flongle and the run initiated with the high-accuracy base calling model (Guppy 5.0.17). The run generated about 429000 reads of which more than 450,000 had a QC > 9. The reads were analyzed with the dedicated EPI2ME 16 S pipeline (v2022.01.07) using the following parameters: minimum score >Q10, minimum length 1000, maximum length 2200, minimum coverage 50%, 5 max target sequence and two different BLAST identity thresholds were tested (85 or 90%). This version used 22,162 sequences of reference coming from the microbial 16 S rRNA NCBI RefSeq database.

## Reporting summary

Further information on research design is available in the Nature Portfolio Reporting Summary linked to this article.

## Data availability

All data, including the source data of the figures, the genotypes, the chemicals used and the detailed statistical analyses including the exact p values are available in the source data file https://doi.org/10.6084/m9.figshare.24972729.v1 Source data are provided with this paper.

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

## Acknowledgements

We thank François Leulier, Sandrine Hughes and Maxime Veites for their help with microbiota sequencing. We thank Emilie Avazéri, Gladys Gazelle and Romane Milleville for their technical help. This work was supported by CNRS, ANR BACNEURODRO (ANR-17-CE16-0023-01), Equipe Fondation pour la Recherche Médicale (EQU201603007783) et l'Institut Universitaire de France to J.R. and the ANR Pepneuron (ANR-21-CE16-0027) to J.R. and Y.G. Research in Y.G.'s laboratory is supported by the CNRS, the "Université de Bourgogne Franche-Comté", the Conseil Régional Bourgogne Franche-Comté (PARI grant), the FEDER (European Funding for Regional Economical Development), and the European Council (ERC starting grant, GliSFCo-311403).

## Author contributions

M.M., G.M., L.K, Y.G. and J.R. conceived the experiments. M.M., G.M., M.B.G., Y.D. and G.G., performed the experiments. M.M., L.K. and J.R. wrote the manuscript. Y.G. and J.R. secured funding.

## Competing interests

The authors declare no competing interests.
