## [Peer Review File · Nature Communications]

REVIEWER COMMENTS

Reviewer #1 -- *Drosophila* biology / genetics / neuro / gustative -- (Remarks to the Author):

The work by Montanari et al. focus on role of the gustatory system in the detection of bacteria. In particular, the authors demonstrated that flies avoid peptidoglycan (PGN) and that this mechanism is mediated by ppk23 and gr66a positive neurons. Interestingly, the ability of adults to avoid PGN depends on the exposure of larvae to bacteria. Overall, the results are interesting, and the methodology used is sound. However, the novelty of the study is limited compared to the existing knowledge on flies ability to detect bacterial components. The strongest novelty of this study is the identification of a developmental time window during which the larvae need to be exposed to bacteria to give rise to competent adults. Following, some points that need to be addressed:

1. From line 142 the authors say to have used an intersectional genetic strategy to quantify calcium activity modification only in ppk23+/gr66A- cells. I think it would be important and make the text clearer to add a brief description of the methodology used. Moreover, it is inappropriate to refer to a figure S8 at this point in the paper, where the previously cited supplementary figure is S2. The numbering of the supplemental figures should be sequential.

2. The authors claim that the ppk23+/gr66a- cells are sufficient to mediate PNG-induced PER suppression (Line 143). But I think their genetic experiments do not support this claim. In fact, in figure S8A, when the ppk23+/gr66a- cells are repressed the flies remain able to sense and avoid PNG, so this proves that they are not necessary for the behavior, that can be sustained by gr66a+ cells. In line with this, in the S8 legend (line 681) the authors state "Flies in which Gr66a-/ppk23+ cells are silenced remains able to sense PGN." On the other hand, the silencing of all gr66a+ or all the ppk23+ neurons leads to loss of PNG-induced PER suppression, suggesting that both gr66a+/ppk23+ neurons are required for the aversion (Figure 2A,B). If the ppk23+/gr66a- cells would be sufficient to mediate the PNG avoidance, then in presence of silencing of gr66a+ cells, the flies should still present this behavior, which is not the case. The fact that ppk23+/gr66a- neurons do respond to PNG (Figure 2C,D) does not mean they are sufficient to mediate the behavior.

The same comment is related to line 281/282, where the authors claim that "gr66a cells are dispensable and ppk23+ cells essential". Are the authors referring to Gr66a+/ppk23- cells when talking about dispensable cells? Because in this case it should be stated in the text for better clarity.

3. In line 146-147 the authors say that "these cells did not respond to caffeine but did to high salt" but there is no reference to a figure or a panel within a figure.

Minor points:

1. In the figure legend is indicated "Further details can be found in the detailed lines, conditions and, statistics for the figure section". It is not clear to me to which sections they are referring to.
2. In figure 1A the authors claim that "In PER, Dap-PGN was found to be aversive to reference flies (w-) when used at 100 (PGN100) and 200 micrograms/mL (PGN200) (Lines 104-105). However, in the graph of Figure 1A no asterisks pointing to significance are indicated between the sucrose bar and the E.C 100 bar.
3. In figure S4A the Gr66aGal4;UAS-pgrp-LERNAi show reduction but no significant difference between sucrose and sucrose+PNG. Since this is the only RNAi that does not show significant difference, could the authors comment?
4. In line 197 it is indicated that "the number of bacteria by fly was strongly reduced", how is this measured? I saw the detailed description in the methods section, but I think it should be briefly cited in the text.
5. In figure 3A legend, line 465 "flies are shifted from 18°C (green) to 29°C (blue)" but actually in the figure the 29°C is represented in red.

Reviewer #2 -- *Drosophila* microbiome / development / behavior -- (Remarks to the Author):

The paper being reviewed explores the mechanisms that drive the gustatory aversive response to peptidoglycan (PGN) in *Drosophila melanogaster* at the neuronal and molecular level. Using a mix of behavioral and neuronal imaging approaches, the study reveals that two types of neurons, ppk23+ and Gr66a+, mediate the aversive response to PGN. The authors found that ppk23+ neurons are required to detect PGN only in the adult stage, while Gr66a+ neurons are required during the larval stage to prime the animal to recognize PGN later in adulthood. The study also shows that the presence of commensal bacteria during the larval stage is necessary for the animal to exhibit an aversive response to PGN in adulthood, as animals raised in germ-free conditions do not show PGN aversion. Finally, the authors demonstrate that the association of axenic larvae with commensal bacteria *L. brevis* is sufficient to restore the PGN aversive response, which is not observed when axenic adults are associated with the bacteria.

Overall, the authors present compelling findings supported by relevant experiments. However, some additional experiments are necessary to solidify the conclusions and explicitly test the presented model and predictions. At this stage many of the proposed interesting insights are indirect conclusions and would profit from being explicitly tested. Furthermore, in some points the text lacks clarity and the description of relevant technical details to interpret the findings. Finally the authors fail to mention that Jia et al. (Nature comm. 2021) have already shown that larval exposure to

commensal bacteria is an important factor priming the behavior of adult flies (aggression). While the paper is cited the authors claim that this is the first study describing this phenomenon. Something which is clearly not the case.

Major Comments:

All conclusions from the authors are based on the assumption that the Gal4 lines being used manipulate specific neurons. This is however never tested. Therefore the authors should characterize and demonstrate that the used Gal4 driver line expression patterns indeed label the desired neurons and that the labelling is specific to these (do not label other cells). They should do so both in the larva and the adult. This is key to confirm the specificity of the manipulations and therefore also conclusions. Solely relying on citing the literature is not sufficient.

It is often not clear why the authors decide to do specific statistical comparisons to reach their conclusions. In general the statistical comparison should be done to the relevant control situation. For example in line 253, the authors claim that “When the same experiment was performed with a related species, *Lactobacillus plantarum*, no PER inhibition was observed (Fig 7C).” this is contradictory to the significant effect shown in Fig. 7C when compared to MRS media control. Which is the relevant control. This is an issue with many statistical tests and comparisons. Also when comparing with genetic controls (e.g. Fig 4C, Fig. S4A etc.). This should be addressed as it affects the conclusions in multiple parts of the manuscript. Along these lines, in several cases the authors refer to the supplemental table for statistical information. “Further details can be found in the detailed lines, conditions and, statistics for the figure section.” The information in this file (for example how n was calculated) is difficult to understand. The authors should annotate this information.

Along these lines, it is not clear if the authors correct for multiple comparisons in their PER analysis? This would be important when comparing multiple conditions. Furthermore, the authors plot open circles over all of their bar plots but fail to explain their meaning. It would be important to add an explanation of their meaning and how they relate to the statistics that were used.

One of the key assumptions of the manuscript is that larval Gr66a+ neurons are activated in the larva. In order to show that in fact larval Gr66a+ neurons are activated the authors should image Gr66a+ neurons in for example axenic vs conventional larvae to support this indirect conclusion. Protocols to do that exist and should be more straightforward than adult imaging.

Likewise the idea that activity of specific larval neurons “prime” the activity pattern of adult neurons and the corresponding behavior is something which can and should be tested directly. The authors could use thermogenetics or optogenetics to complement the silencing experiments and test if activity in these neurons is sufficient to generate the adult phenotype.

The authors nicely show that RNAi knockdown of TRPA1 in larval Gr66a+ neurons abolishes the aversive response of adult flies towards PGN. The presented data are behavioural. An obvious conclusion is that this phenotype arises from adult Ppk23+ neurons not being sensitive to PGN anymore. It would be important to test this by imaging Ppk23+ neurons in the adults in which TRPA1 was knocked down at the larval stage in Gr66a+ neurons. This experiment might be genetically tricky but if somehow feasible experiments along these lines would strongly support the presented conclusions.

In general it is very difficult to assess the quality of the imaging data. This is due to issues with how the data are presented and the description of the experiments in the methods section. These are the several issues with how the imaging data are presented:

- The scale bar of the traces shown are not annotated, so the strength of the signal cannot be determined.
- The authors don't mention how the stimulus was delivered.
- The authors don't mention the gustatory stimulation protocol.
- The gustatory stimulant used for the imaging is not the same as for the behaviour (it does not seem to contain sucrose). That would be key to relate the imaging experiments to the behavioural experiments (where the stimulus contains sucrose).
- It is not clear what the n of the experiment was or how it was calculated. The sentence in the methods section is confusing: "A minimum of 2 independent experiments with a total n for each condition ranging from 7 to 9 were performed."
- The authors refer to a paper in the methods section but the citation is (Ref).
- For some imaging experiments (E.g. in Fig S5) the relevant control (e.g. is missing). It is therefore not possible to say if there is a relevant activation of the neurons upon stimulation.
- In line 147 the authors say that the cells respond to high salt specifically. It would be important to point the reader to the relevant panel. This is an important panel.

The authors show that Gr66a+ neurons are activated by PGN in both aseptic and non-aseptic adults (Fig. S5F). As activating these bitter neurons normally conveys an aversive response, it is surprising that in their hands this activation is not sufficient to show a suppression of PER (Fig. 3b – flies do not show aversion in a situation where only ppk23+ neurons are silenced in the adult). Could the authors please elaborate on this?

In general the absence of phenotype when manipulating the IMD pathway should be taken with a grain of salt as there is no evidence that the knockdown in the targeted neurons works. If one wants to make a negative conclusion (absence of phenotype) one would need to provide evidence for the

knockdown (for example using qRT-PCR or antibodies). Also, given that the gram-positive bacteria *L. brevis* restores the PGN aversive response (Fig. 4 & S4) shouldn't one consider that possibly the receptors activated in Gr66a+ neurons (in larvae) or in ppk23+ neurons (in adults) that mediate this effect are from the Toll pathway instead of the IMD pathway? While not strictly required it would be relevant that the authors test the effect of silencing elements of the Toll pathway (eg. PGRP-SA, Spz).

The manuscript is interesting and relevant for multiple reasons including the mechanistic dissection of the gustatory effects of commensal bacterial exposure. But at this stage their claim of novelty is overblown: The authors' claim in line 324 that "If recent reports describe the influence that microbiota has on certain adult fly behaviors 60, 62, none of them report on the bacterial requirement specifically in larvae" is incorrect, as cited reference 62 specifically demonstrates that bacterial exposure during the larval stage (48-96h post oviposition) is necessary for aggressive behavior (Figure 5f). This study and their finding should be given more extensive space in the introduction, results section, and discussion. Similarities and differences should be compared between both studies. Also, the authors demonstrate that knocking down TRPA1 in larval Gr66a+ neurons abolishes the aversive response of adult flies towards PGN (Fig. 4c). While discussing the role of TRPA1 in aversive taste stimulation, the authors cite Soldano et al (52, 53), but do not acknowledge that this paper has already implicated TRPA1 channels in Gr66a+ neurons in the detection of bacterial components such as LPS. This should be made more explicit. In general, giving a bit more space to the discussion of the effects of bacteria (both their detection and also the effect of commensal and pathological bacteria colonialization) on gustatory mediated behaviors in flies would be pertinent.

Minor Comments

The inclusion and discussion of data in supplementary Figure 8 in the discussion is very confusing. It is very difficult to follow the presented data and conclusions. Also, I would suggest to not include data in the discussion section.

Throughout the paper, to test the aversive response, the authors have used DAP-type PGN extracted from the gram-negative bacteria *E.coli*. Interestingly the authors show that the gram-positive *L. brevis* rescues the PGN aversive response. However, given the nature of the PGN used, the authors could also test if other gram-negative species identified in the sequenced microbiome can also rescue the aversive behaviour (*Acetobacter* species for example). While not being strictly necessary this would be an interesting and pertinent experiment.

The relevant genetic controls are often missing (e.g. for the octopamine mutants).

The authors use a surprisingly low concentration of sucrose to perform the PER experiments. This is discussed in the methods section. But maybe it would be important to mention this in the results section as this is relevant for putting the relevance of the results in context for a superficial and non-expert reader. I assume this means the aversive response is not very strong.

The authors consistently use the format XXGal4 to identify Gal4 lines throughout the manuscript, which may be confusing to readers as it suggests the use of specific alleles of the indicated genes instead of the use of Gal4 transgenes.

Currently, the authors extensively discuss potential mechanisms but fail to discuss possible physiological relevance of the observed phenomenon. The authors could rebalance the discussion by making the mechanistic discussion more concise and add a discussion section about potential ecological relevance of an adaptive response to bacterial components.

Overall the paper is well structured. However, some sentences have some language issues and are therefore difficult to understand. For instances sentences contained in lines:

- In general substitute expression “larvae give birth to flies” with “flies hatched from larvae”;
- Lines 146 to 147 - the Figure where the result described was observed need to be stated;
- Line 188- “TRPA1 receptor” should be “TRPA1 channel”;
- Line 194- instead of “burrow” use “born”;
- Line 197- instead of “by” use “per”;
- Line 215- instead of “form” use “from”;
- Line 217- instead of “non-aseptic” use “conventional”;
- Line 217 to 219 – The structure and the message in the sentence is not clear.
- Line 226 – The exact larval stage at which larvae are changed to a new tube is not explicit anywhere in the text. Similarly, it is never mentioned in methods, figures or figures legends.
- Line 348 to 351 - The sentence is not clear, particularly the use of the word “competence”.
- line 769: “The gut was cut without damaging the brain and taste nerves to allow visual 769 access to the anterior ventral part of the sub-esophageal zone (SEZ).” We assume it was not the gut but the esophagus that was cut.
- In general it is not correct to describe the aversive effect of compounds as “triggering PER” (line 179 for example).

- The paragraph starting at line 137 is generally difficult to understand. Also it is unclear why the authors suddenly focus on these specific neurons. Same for the following section.

Fig 3. In general, the fly cycle scheme used repeatedly throughout the paper is poorly described and perhaps even misleading at some point. The authors chose to represent two adult flies in tandem with different body colours. Are these young and adult flies? Is it related to their genotype? Is it a transformation of light-colour flies into dark-flies? Also, the experiments never use the offspring and therefore the cycle is not necessarily required. A linear scheme might be more appropriate?

Fig. 3A “blue” should be “red”. It is never mentioned at what stage the animals are moved to a permissive/restrictive temperature. As pupae? After adults hatched?

Fig 6. In panel A, two genotypes are used to distinguish conventional and axenic flies. Unfortunately, the same colour code was used to highlight the septic state of flies that is used in the life cycle scheme. Later this overlap makes the interpretation of Fig S6 confusing. In panel B, the flies that come from axenic (w-) larvae should have lighter than those coming from conventional (yw-) flies.

Fig S5F – The response to water should be plotted (negative control), and the statistical comparison of water with PGN should be added to the figure.

Fig S8. Figure numbering does not match the order in the text. Fig S8 is mentioned after Fig S1.

Fig S8 – This experiment lacks positive and negative controls. For example, +/UAS-kir2.1 and Gr66a+Gal4/UAS-Kir2.1 (which abolishes PGN aversive response).

Figure S8A – caption and Figure: the annotation ppk23Glut for the Gal80 intersection seems misleading “ppk23Glut’ indicates ppk23Gal4 with Gr66aLexA and LexAop685 Gal80; and ‘>’ denotes Gal4 driving UAS-Kir2.1 with temporal restriction by tub-Gal80ts.”

The authors do not describe how and why the diet was manipulated to manipulate the protein and carbohydrate content of the diet.

Conclusion:

We congratulate the authors for presenting a stimulating and relevant phenomenon and exploring it mechanistically. They discover many interesting aspects of possible mechanisms which however are not always strongly supported. There are multiple experiments and improvements to the analysis of the experiments which are required to validate the robustness of the presented data and also to directly validate the indirectly drawn conclusions. Due to this we are not able to support the publication of the manuscript in its current form and recommend that the authors get the opportunity to revise their manuscript. This should allow them to provide strong data to support their interesting claims.

Dear editors and reviewers,

We would like to thank the reviewers for their detailed and thorough analysis of our manuscript. Many of the questions they asked correspond to questions we are also asking ourselves. We have tried to address their concerns as best as we can and we believe that the manuscript has been improved at this stage of the review. Below is a point-by-point response

REVIEWER COMMENTS

Reviewer #1 -- Drosophila biology / genetics / neuro / gustative -- (Remarks to the Author):

The work by Montanari et al. focus on role of the gustatory system in the detection of bacteria. In particular, the authors demonstrated that flies avoid peptidoglycan (PGN) and that this mechanism is mediated by ppk23 and gr66a positive neurons. Interestingly, the ability of adults to avoid PGN depends on the exposure of larvae to bacteria. Overall, the results are interesting, and the methodology used is sound. However, the novelty of the study is limited compared to the existing knowledge on flies ability to detect bacterial components. The strongest novelty of this study is the identification of a developmental time window during which the larvae need to be exposed to bacteria to give rise to competent adults. Following, some points that need to be addressed:

1.1 1. From line 142 the authors say to have used an intersectional genetic strategy to quantify calcium activity modification only in ppk23+/gr66A- cells. I think it would be important and make the text clearer to add a brief description of the methodology used.

We have added this paragraph.

This genetic strategy is based on the co-expression of specific genes in cell populations. Based on the literature, three subpopulations of taste neurons could exist: those expressing Gr66a but not ppk23 (Gr66a+ ppk23-), those co-expressing Gr66a and ppk23 (Gr66a+ ppk23+) and those expressing ppk23 but not Gr66a (ppk23+ Gr66a-). When genetically combined, the ppk23-Gal4 and GR66a-Gal80 transgenes gave rise to flies in which the Gal4 driver is expressed only in ppk23+Gr66a- cells.

1.2 Moreover, it is inappropriate to refer to a figure S8 at this point in the paper, where the previously cited supplementary figure is S2. The numbering of the supplemental figures should be sequential.

The Sup figures have been modified and the numbering of the supplemental figures is now sequential

1.3. The authors claim that the ppk23+/gr66a- cells are sufficient to mediate PNG-induced PER suppression (Line 143). But I think their genetic experiments do not support this claim. In fact, in figure S8A, when the ppk23+/gr66a- cells are repressed the flies remain able to sense and avoid PNG, so this proves that they are not necessary for the behavior, that can be sustained by gr66a+ cells. In line with this, in the S8 legend (line 681) the authors state "Flies in which Gr66a-/ppk23+ cells are silenced remains able to sense PGN." On the other hand, the silencing of all gr66a+ or all the ppk23+ neurons leads to loss of PNG-induced PER suppression, suggesting that both gr66a+/ppk23+ neurons are required for the aversion (Figure 2A,B). If the ppk23+/gr66a- cells would be sufficient to mediate the PNG avoidance, then in presence of silencing of gr66a+ cells, the flies should still present this behavior, which is not the case.

When we inactivate all the Gr66a or all the ppk23 cells, we inactivate them throughout the life of the fly. Larvae and adults. The temporal inactivation of Gr66a+ cells only in adults demonstrates that, at the adult stage, the Gr66a+ population, which includes Gr66a+ ppk23+, is not required to respond to PGN.

The fact that ppk23+/gr66a- neurons do respond to PNG (Figure 2C,D) does not mean they are sufficient to mediate the behavior.

The same comment is related to line 281/282, where the authors claim that "gr66a cells are dispensable and ppk23+ cells essential". Are the authors referring to Gr66a+/ppk23- cells when talking about dispensable cells? Because in this case it should be stated in the text for better clarity.

To make this clear, we have added the word "adult" to the following sentence.

Therefore, some sensilla host both Gr66a+/ppk23+ and Gr66a-/ppk23+ cells. Since our data show that adult Gr66a+ neurons are dispensable and adult ppk23+ cells essential for PGN-mediated PER, we could conclude that Gr66a-/ppk23+ cells are the only cells involved

1.4. In line 146-147 the authors say that “these cells did not respond to caffeine but did to high salt” but there is no reference to a figure or a panel within a figure.

This had been added (Fig 2E)

Minor

points:

1.5. In the figure legend is indicated “Further details can be found in the detailed lines, conditions and statistics for the figure section”. It is not clear to me which sections they are referring to.

We provided an Excel file with all the raw data and statistics used. We have included a paragraph for Statistics.

1.6. In figure 1A the authors claim that “In PER, Dap-PGN was found to be aversive to reference flies (w-) when used at 100 (PGN100) and 200 micrograms/mL (PGN200) (Lines 104-105). However, in the graph of Figure 1A no asterisks pointing to significance are indicated between the sucrose bar and the E.C 100 bar.

It was an oversight that has been corrected.

1.7. In figure S4A the Gr66aGal4;UAS-pgrp-LERNAi show reduction but no significant difference between sucrose and sucrose+PNG. Since this is the only RNAi that does not show significant difference, could the authors comment?

We have increased the number of flies tested and the difference is now statistically significant.

1.8. In line 197 it is indicated that “the number of bacteria by fly was strongly reduced”, how is this measured? I saw the detailed description in the methods section, but I think it should be briefly cited in the text.

We have added “by CFU plating”.

1.9. In figure 3A legend, line 465 “flies are shifted from 18°C (green) to 29°C (blue)” but actually in the figure the 29°C is represented in red.

It was an unfortunate error that we have corrected

Reviewer #2 -- Drosophila microbiome / development / behavior -- (Remarks to the Author):

The paper being reviewed explores the mechanisms that drive the gustatory aversive response to peptidoglycan (PGN) in *Drosophila melanogaster* at the neuronal and molecular level. Using a mix of behavioral and neuronal imaging approaches, the study reveals that two types of neurons, ppk23+ and Gr66a+, mediate the aversive response to PGN. The authors found that ppk23+ neurons are required to detect PGN only in the adult stage, while Gr66a+ neurons are required during the larval state to prime the animal to recognize PGN later in adulthood. The study also shows that the presence of commensal bacteria during the larval stage is necessary for the animal to exhibit an aversive response to PGN in adulthood, as animals raised in germ-free conditions do not show PGN aversion. Finally, the authors demonstrate that the association of axenic larvae with commensal bacteria *L. brevis* is sufficient to restore the PGN aversive response, which is not observed when axenic adults are associated with the bacteria.

Overall, the authors present compelling findings supported by relevant experiments. However, some additional experiments are necessary to solidify the conclusions and explicitly test the presented model and predictions. At this stage many of the proposed interesting insights are indirect conclusions and would profit from being explicitly tested. Furthermore, in some points the text lacks clarity and the description of relevant technical details to interpret the findings. Finally the authors fail to mention that Jia et al. (Nature comm. 2021) have already shown that larval exposure to commensal bacteria is an important factor priming the behavior of adult flies (aggression). While the paper is cited the authors claim that this is the first study describing this phenomenon. Something which is clearly not the case.

It was not our intention to downplay the interesting work of Jia et al. Our work demonstrates once again the importance of gut-associated microbes on neural-controlled behavior in adults. We have added a paragraph to the discussion to better cite this reference and make comparisons with our study.

Major

Comments:

2.1 All conclusions from the authors are based on the assumption that the Gal4 lines being used manipulate specific neurons. This is however never tested. Therefore the authors should characterize and demonstrate that the used Gal4 driver line expression patterns indeed label the desired neurons and that the labelling is specific to these (do not label other cells). They should do so both in the larva and the adult. This is key to confirm the

specificity of the manipulations and therefore also conclusions. Solely relying on citing the literature is not sufficient.

As requested by this reviewer, we carried out extensive mapping of cells expressing Gal4 protein in larvae and adults of ppk23Gal4 and Gr66aGal4 flies. The results are now presented in Figure S3, S4 and S5. The patterns obtained are consistent with those reported in the literature and confirm all the functional tests we had performed with these drivers, which both behaved as described in the literature. These patterns will be useful when trying to map more precisely the Gr66a cells involved in the priming step.

2.2 It is often not clear why the authors decide to do specific statistical comparisons to reach their conclusions. In general, the statistical comparison should be done to the relevant control situation. For example, in line 253, the authors claim that “When the same experiment was performed with a related species, *Lactobacillus plantarum*, no PER inhibition was observed (Fig 7C).” this is contradictory to the significant effect shown in Fig. 7C when compared to MRS media control. Which is the relevant control. This is an issue with many statistical tests and comparisons. Also, when comparing with genetic controls (e.g. Fig 4C, Fig. S4A etc.). This should be addressed as it affects the conclusions in multiple parts of the manuscript.

This has been changed in the figures and the statistical comparisons now only rely on the relevant control situation. We have added a “Statistic” paragraph.

Along these lines, in several cases the authors refer to the supplemental table for statistical information. “Further details can be found in the detailed lines, conditions and, statistics for the figure section.” The information in this file (for example how n was calculated) is difficult to understand. The authors should annotate this information. This has been done, tables with the number of flies tested in each experiment have been added to the supplementary file.

Along these lines, it is not clear if the authors correct for multiple comparisons in their PER analysis? This would be important when comparing multiple conditions.

The Fisher's exact test does not directly account for multiple comparisons. It is designed to test the association between two specific categorical variables (PER or No PER and test vs Control), and the p-value it produces reflects the strength of evidence against the null hypothesis of independence between those variables. All our conclusions are obtained from the comparison of the relevant control with one test and not from the analysis of all the tests at once as our only hypothesis is a possible difference between the tested situation and the control. Therefore, we did not correct for multiple comparisons.

Furthermore, the authors plot open circles over all of their bar plots but fail to explain their meaning. It would be important to add an explanation of their meaning and how they relate to the statistics that were used.

Open circles represent the average PER value for an independent experiment. The bar is the mean obtained from all the flies, from all the independent experiments. A Fisher Exact Test is applied to compare the decisions from all the flies from the relevant control with the flies from the test.

2.3 One of the key assumptions of the manuscript is that larval Gr66a+ neurons are activated in the larva. In order to show that in fact larval Gr66a+ neurons are activated the authors should image Gr66a+ neurons in for example axenic vs conventional larvae to support this indirect conclusion. Protocols to do that exist and should be more straightforward than adult imaging.

This is indeed an important experiment to try, However, since it is still unclear when and how exactly are Gr66a cells activated by microbiota (which are constantly present in the gut), we choose to test whether an acute “stimulation” of Gr66a cells by *L. brevis* would activate Gr66a neurons. And indeed, when larval heads are incubated with solution contaminated with *L. brevis*, this is sufficient to induce increased calcium levels in Gr66a cells. These new data are presented Fig 7E and 7F.

Likewise, the idea that activity of specific larval neurons “prime” the activity pattern of adult neurons and the corresponding behavior is something which can and should be tested directly. The authors could use thermogenetics or optogenetics to complement the silencing experiments and test if activity in these neurons is sufficient to generate the adult phenotype.

This is another interesting experiment to test whether the activation of larval Gr66a cells would be sufficient to trigger priming. To do this, we used UAS-TRPA1ts, which allows us to constitutively activate Gr66a cells independently of any ligand. As we show below, activation of Gr66a by TRPA1 in axenic larvae was not sufficient to render adults capable of responding to PGN. The simplest conclusion is that larval Gr66a activation is not

sufficient to induce priming. But as this is a negative result, it could be interpreted in many other ways. Much more work will be needed to resolve this issue.

D

2.4 The authors nicely show that RNAi knockdown of TRPA1 in larval Gr66a+ neurons abolishes the aversive response of adult flies towards PGN. The presented data are behavioural. An obvious conclusion is that this phenotype arises from adult Ppk23+ neurons not being sensitive to PGN anymore. It would be important to test this by imaging Ppk23+ neurons in the adults in which TRPA1 was knocked down at the larval stage in Gr66a+ neurons. This experiment might be genetically tricky but if somehow feasible experiments along these lines would strongly support the presented conclusions.

To do this experiment, we need to reduce the TRPA1 mRNA levels in the larvae in a temporal manner, using the Gal4/Gal80ts system. At the same time, we need to express the GCaMP in ppk23 glut neurons which means in ppk23+Gr66a- neurons. This later also depends on the Gal4 system making the experiment not feasible with the tools we have.

2.5 In general it is very difficult to assess the quality of the imaging data. This is due to issues with how the data

are presented and the description of the experiments in the methods section. These are the several issues with how the imaging data are presented:

- The scale bar of the traces shown are not annotated, so the strength of the signal cannot be determined.

This has been added

- The authors don't mention how the stimulus was delivered.

This has been done

- The authors don't mention the gustatory stimulation protocol.

This has been done

- The gustatory stimulant used for the imaging is not the same as for the behaviour (it does not seem to contain sucrose). That would be key to relate the imaging experiments to the behavioural experiments (where the stimulus contains sucrose).

The sucrose is mandatory for the PER assays in order to delineate whether there is an aversion or not. However, the in vivo calcium imaging is dedicated to assaying whether cells respond to a specific chemical. Therefore, the addition of sucrose during the assays may lead to confounding effects.

- It is not clear what the n of the experiment was or how it was calculated. The sentence in the methods section is confusing: "A minimum of 2 independent experiments with a total n for each condition ranging from 7 to 9 were performed."

This has been done

- The authors refer to a paper in the methods section but the citation is (Ref).

This has been corrected

- For some imaging experiments (E.g. in Fig S5) the relevant control (e.g. is missing). It is therefore not possible to say if there is a relevant activation of the neurons upon stimulation.

An ad hoc control has been added

- In line 147 the authors say that the cells respond to high salt specifically. It would be important to point the reader to the relevant panel. This is an important panel.

An ad hoc control has been added (Fig. 2E)

2.6 The authors show that Gr66a+ neurons are activated by PGN in both aseptic and non-aseptic adults (Fig. S5F). As activating these bitter neurons normally conveys an aversive response, it is surprising that in their hands this activation is not sufficient to show a suppression of PER (Fig. 3b – flies do not show aversion in a situation where only ppk23+ neurons are silenced in the adult). Could the authors please elaborate on this?

From the GCaMP analysis, it is clear that calcium increases in Gr66a+ cells after exposure to PGN, but although we can conclude that the cells respond to the chemical, we cannot say that the cell is activated or fully activated, leading to a phenotype such as PER. Indeed, calcium levels are lower in PGN-stimulated Gr66a cells than those obtained after exposure to caffeine. It is conceivable that a certain level of activation is required to trigger PER. Such a level is reached with caffeine but not with PGN.

2.7 In general the absence of phenotype when manipulating the IMD pathway should be taken with a grain of salt as there is no evidence that the knockdown in the targeted neurons works. If one wants to make a negative conclusion (absence of phenotype) one would need to provide evidence for the knockdown (for example using qRT-PCR or antibodies).

These RNAi lines against IMD pathway components have been used in many reports and shown to block for example the production of IMD dependent antimicrobial peptides after bacterial infection. It is however difficult to make sure that they are efficient in these specific neurons. To our knowledge there are no antibodies against IMD pathway components that have been validated that will allow us to test if RNAi-mediated downregulation works here. But it should be mentioned that we have recently shown that some of these lines (but not all) blocks PGN mediated signal in Gr66a cells. [10.1523/JNEUROSCI.2437-21.2022](https://doi.org/10.1523/JNEUROSCI.2437-21.2022).

2.8 Also, given that the gram-positive bacteria *L. brevis* restores the PGN aversive response (Fig. 4 & S4) shouldn't one consider that possibly the receptors activated in Gr66a+ neurons (in larvae) or in ppk23+ neurons (in adults)

that mediate this effect are from the Toll pathway instead of the IMD pathway? While not strictly required it would be relevant that the authors test the effect of silencing elements of the Toll pathway (eg. PGRP-SA, Spz). We have analyzed the putative implication of the Toll pathway by testing the effect of MyD88 inactivation on PER. The new data presented in Fig4C show that it has no effect.

2.9 The manuscript is interesting and relevant for multiple reasons including the mechanistic dissection of the gustatory effects of commensal bacterial exposure. But at this stage their claim of novelty is overblown: The authors' claim in line 324 that "If recent reports describe the influence that microbiota has on certain adult fly behaviors 60, 62, none of them report on the bacterial requirement specifically in larvae" is incorrect, as cited reference 62 specifically demonstrates that bacterial exposure during the larval stage (48-96h post oviposition) is necessary for aggressive behavior (Figure 5f). This study and their finding should be given more extensive space in the introduction, results section, and discussion. Similarities and differences should be compared between both studies.

It was not our intention to downplay the interesting work of Jia et al. Our work demonstrates once again the importance of gut-associated microbes on neural-controlled behavior in adults. We have added a paragraph to the discussion to better cite this reference and make comparisons with our study.

Also, the authors demonstrate that knocking down TRPA1 in larval Gr66a+ neurons abolishes the aversive response of adult flies towards PGN (Fig. 4c). While discussing the role of TRPA1 in aversive taste stimulation, the authors cite Soldano et al (52, 53), but do not acknowledge that this paper has already implicated TRPA1 channels in Gr66a+ neurons in the detection of bacterial components such as LPS. This should be made more explicit. In general, giving a bit more space to the discussion of the effects of bacteria (both their detection and also the effect of commensal and pathological bacteria colonization) on gustatory mediated behaviors in flies would be pertinent.

This has been done in the discussion

Minor

Comments

2.10 The inclusion and discussion of data in supplementary Figure 8 in the discussion is very confusing. It is very difficult to follow the presented data and conclusions. Also, I would suggest to not include data in the discussion section.

This has been changed

2.11 Throughout the paper, to test the aversive response, the authors have used DAP-type PGN extracted from the gram-negative bacteria E.coli. Interestingly the authors showed that the gram-positive L. brevis rescues the PGN aversive response. However, given the nature of the PGN used, the authors could also test if other gram-negative species identified in the sequenced microbiome can also rescue the aversive behaviour (Acetobacter species for example). While not being strictly necessary this would be an interesting and pertinent experiment. We agreed with the reviewer that other bacteria species need to be tested. This would be very useful especially to identify the signal that mediates the priming. But this should be done on a large-scale basis to have a real meaning. Recolonization is not so easy since the ability of the bacteria to survive is dependent on the fly' diet. We might need to change it, which includes many controls. This would be another study.

2.12 The relevant genetic controls are often missing (e.g. for the octopamine mutants).

They have been added

2.13 The authors use a surprisingly low concentration of sucrose to perform the PER experiments. This is discussed in the methods section. But maybe it would be important to mention this in the results section as this is relevant for putting the relevance of the results in context for a superficial and non-expert reader. I assume this means the aversive response is not very strong.

The aversive response is strong, with 20% of the flies attracted to sucrose when it is mixed with PGN, compared to 70% when it was alone. The aversion is in fact as strong as with quinine or caffeine if we refer to the % of flies rejecting the solution. For caffeine too, increasing concentrations of sucrose reduces aversion. Fig S1C We had a sentence in the text.

2.14 The authors consistently use the format XXGal4 to identify Gal4 lines throughout the manuscript, which

may be confusing to readers as it suggests the use of specific alleles of the indicated genes instead of the use of Gal4 transgenes.

This was a mistake that was corrected

2.15 Currently, the authors extensively discuss potential mechanisms but fail to discuss possible physiological relevance of the observed phenomenon. The authors could rebalance the discussion by making the mechanistic discussion more concise and add a discussion section about potential ecological relevance of an adaptive response to bacterial components.

A paragraph has been added in the beginning of the discussion

2.16 Overall the paper is well structured. However, some sentences have some language issues and are therefore difficult to understand. For instances sentences contained in lines:

- In general substitute expression “larvae give birth to flies” with “flies hatched from larvae”;

Done

- Lines 146 to 147 - the Figure where the result described was observed need to be stated;

Done

- Line 188- “TRPA1 receptor” should be “TRPA1 channel”;

Done

- Line 194- instead of “burrow” use “born”;

Done

- Line 197- instead of “by” use “per”;

Done

- Line 215- instead of “form” use “from”;

Done

- Line 217- instead of “non-aseptic” use “conventional”;

Done

- Line 217 to 219 – The structure and the message in the sentence is not clear.

This has been changed

- Line 226 – The exact larval stage at which larvae are changed to a new tube is not explicit anywhere in the text. Similarly, it is never mentioned in methods, figures or figures legends.

This has been done in the discussion

- Line 348 to 351 - The sentence is not clear, particularly the use of the word “competence”.

This has been changed

- line 769: “The gut was cut without damaging the brain and taste nerves to allow visual access to the anterior ventral part of the sub-esophageal zone (SEZ).” We assume it was not the gut but the esophagus that was cut.

Yes.

- In general it is not correct to describe the aversive effect of compounds as “triggering PER” (line 179 for example).

We agreed. This has been changed.

- The paragraph starting at line 137 is generally difficult to understand. Also it is unclear why the authors suddenly focus on these specific neurons. Same for the following section.

We do not understand this comment. It has been shown that both Gr66a and ppk23 neurons mediate aversive response to bitter compounds in flies. This is the reason why we focus on them.

2.17 Fig 3. In general, the fly cycle scheme used repeatedly throughout the paper is poorly described and perhaps even misleading at some point. The authors chose to represent two adult flies in tandem with different body colours. Are these young and adult flies? Is it related to their genotype? Is it a transformation of light-colour flies into dark-flies? Also, the experiments never use the offspring and therefore the cycle is not necessarily required. A linear scheme might be more appropriate?

This was a mistake. We apologize for that. This has been changed. We agreed with the reviewer that the cycle is confusing since no offspring is involved. We have transformed the circles into lines.

2.18 Fig. 3A “blue” should be “red”. It is never mentioned at what stage the animals are moved to a permissive/restrictive temperature. As pupae? After adults hatched?

This has been changed

2.19 Fig 6. In panel A, two genotypes are used to distinguish conventional and axenic flies. Unfortunately, the same colour code was used to highlight the septic state of flies that is used in the life cycle scheme. Later this overlap makes the interpretation of Fig S6 confusing. In panel B, the flies that come from axenic (w-) larvae should have lighter than those coming from conventional (yw-) flies.

This was a mistake. We apologize for that. This has been changed

2.20 Fig S5F – The response to water should be plotted (negative control), and the statistical comparison of water with PGN should be added to the figure.

This has been done (Fig S9E)

2.21 Fig S8. Figure numbering does not match the order in the text. Fig S8 is mentioned after Fig S1.

This has been modified.

2.22 Figure S8A – caption and Figure: the annotation ppk23Glut for the Gal80 intersection seems misleading “'ppk23Glut' indicates ppk23Gal4 with Gr66aLexA and LexAop685 Gal80; and '>' denotes Gal4 driving UAS-Kir2.1 with temporal restriction by tub-Gal80ts.”

This has been changed (Fig.S6).

2.23 The authors do not describe how and why the diet was manipulate to manipulate the protein and carbohydrate content of the diet.

This has been justified

2.24 Fig S8 – This experiment lacks positive and negative controls. For example, +/UAS-kir2.1 and Gr66a+Gal4/UAS-Kir2.1 (which abolishes PGN aversive response).

They have been added.

Conclusion:

We congratulate the authors for presenting a stimulating and relevant phenomenon and exploring it mechanistically. They discover many interesting aspects of possible mechanisms which however are not always strongly supported. There are multiple experiments and improvements to the analysis of the experiments which are required to validate the robustness of the presented data and also to directly validate the indirectly drawn conclusions. Due to this we are not able to support the publication of the manuscript in its current form and recommend that the authors get the opportunity to revise their manuscript. This should allow them to provide strong data to support their interesting claims.

REVIEWER COMMENTS

Reviewer #1 (Remarks to the Author):

The revised version of the manuscript is significantly improved compared to the initial version, the manuscript is relevant and adds interesting new insights into the discussion of the role of bacteria on gustatory mediated behaviors. My major comments have been addressed as well as the minor points have been clarified or corrected. I believe that the new experiments/controls and the more detailed explanation of the different experiments contributed to provide stronger evidence to support the authors' claims.

I have just a few comments that I would like the authors to correct or clarify:

1. I think the reference 38 should be cited again at the end of the sentence in line 33 "PGN signal in these cells", since it is the one that describes the functional requirement of IMD pathway in these cells.

2. I think it is a bit confusing that in the text, line 104-107, the authors refer to the different types of PNG, such as Dap-PNG (PNG100 and PNG 200) and Lys-type PNG but then in figure 1A these compounds are called E.c. 10/100/200 and S.a 100, therefore referring to the bacteria of origin. I think it would be easier to have the same terms to indicate the PNG both in the text and in the Figure.

3. In line 111 referring to figure S1A/B it is mentioned "sugar or AA rich medium", but in the figure the protein rich medium is the first represented. It would be better to keep the same order in the text and in the figure and to give the same name to the media (either AA+ or prot+). Same is true for S1A figure legend, line 625.

4. The paragraph from line 187 and 191 is very important to precisely highlight the conclusions obtained from the experiments aimed at understanding the temporal requirement of the different classes of neurons. I think it should be rephrased more clearly.

5. The experiment performed to test the involvement of the Toll signaling pathway is not sufficient in my opinion to really rule out its involvement. To get to this conclusion more members of the

signaling pathway should be tested but I do not think it is relevant to the current manuscript. If the authors want to keep this experiment in the manuscript, I think they should tune down the interpretation of the results.

6. In line 258 the sentence “Similarly, rearing germ-free eggs on conventional medium for the first 48 hours of development was sufficient to restore PGN responsiveness in adults (Fig 6D)” does not comment on the 48h-96h interval. In the panel it is evident that the flies coming from the 48h-96h interval show no aversion to PNG or at least significantly reduced aversion compared to the CR flies. However, in the schematic line (blue and gray) on the right it is written 48h-96h AEL aversion to PNG. Can please the author comment on this and rephrase it in the text?

7. I think the data of S2A and S2B should be in the same graph and statistical analysis performed between Sucrose and Sucrose+PNG.

8. In my opinion the genotype of the analyzed flies in the Supplementary figure S3, S4, S5 needs to be indicated in the figure.

9. Referring to the manuscript figures in the discussion session will make the overall discussion much easier to follow.

Reviewer #2 (Remarks to the Author):

Overall there was an effort to provide new data for experiments suggested by the reviewers, such as a thorough analysis of the anatomy of the cells labelled by the used Gal4 lines, testing if the Toll pathway might mediate effects described in the manuscript, and testing if the activation of Gr66+ neurons in larvae are sufficient to induce aversive behaviour in adults. However, many important conclusions remain indirect, and many answers were either elusive or several requests for changes in the text were ignored (some even are stated by the authors to have been performed when this was not the case). It was also not trivial to check the edits as not all changes were tracked and especially changes to the figures were not properly documented. This required a lot of extra time and work to be done thoroughly.

A main weakness remains that the intersectional genetic dissection of the neuronal substrate of the aversive effect and priming remains very poor and indirect. The authors nevertheless continue making very strong statements. As described below in more detail, we continue to find their conclusions confusing and often not warranted. Mainly because they are very indirect. Also the statistical tests continue to be problematic and not done across genotypes (with the corresponding controls). While the authors claim they have fixed this or explain why it needs to be done the way they do it, the conclusions they draw requires that they test the effect of for example the suc+Pgn stimulation in the manipulated genotype with the effect of suc+pgn in the control genotypes. This has not been done and it is therefore difficult to make conclusions.

Furthermore the authors dismiss many requests to test their hypothesis directly and when they do so (activating specific neurons in axenic larvae (see specific criticism below) they dismiss the results given that they do not provide the expected outcome. They do not discuss why the results are not trustworthy and simply decide to not include the results in the main manuscript. If accepted we strongly advise to include these negative data so that the reader can make their own conclusions about this experiment which fails to support their model. We made specific comments and recommendations below.

Finally, a challenge is that the details of the experimental protocol still remain unclear (e.g. how long was the gustatory stimulation during the imaging experiments etc.).

In general the phenomenology is interesting and worthy of a thorough mechanistic dissection. The main results describing the phenomenology, should be published as they are valuable for the community. But the current conclusions are very strongly worded and the mechanistic dissection remains rather superficial, indirect, and not very well supported or contradicted by specific results. The revision is a missed opportunity and in our opinion the manuscript should not be published in its current form.

More specifically:

- line 127 - This whole section is (still) very confusing. I think it would help the reader if the authors could explicitly state here, that the manipulations made are effective throughout the life of the animal. for example, by adding something like this:

- "To identify the type of neurons that respond to PGN, PER assays were performed in flies in which 127 specific groups of neurons were inactivated by overexpression of the inward rectifier potassium channel Kir2.1 throughout development."

- line 141- "(ppk23+/Gr66a)" a - is missing form (ppk23+/Gr66a*-*).

- line 145 - "Inactivating the ppk23+/Gr66a- population did not impair the adult aversion toward PGN (Fig S6A). These data suggest that if Gr66a+ and ppk23+ cells are essential to mediate PGN aversion, the ppk23+/Gr66a- subpopulation is dispensable. Consequently, the Gr66a+/ppk23+ cells are implicated in the aversion toward PGN (Fig S6A and S6B)."

- The authors cannot exclude that Gr66a+/ppk23- play a role here as well and should state this here.

- line 154 - a word seems missing in the sentence:

- "Since adult Gr66a+ neurons can respond to PGN, that some labellar Gr66a+ cells are also ppk23+ (Fig S6A and S6B) we used the intersectional genetic strategy described above to quantify calcium activity modifications only in ppk23+/Gr66a- cells 48."

- line 154 – It is not clear what is the logic and relevance of imaging ppk23+/Gr66a- neurons if these neurons were not necessary for behavior? Also not how this finding is relevant for the conclusions.

- line 187ff - "Furthermore, in combination with our data described above indicating that Gr66a+/ppk23+ are sufficient and the demonstration that Gr66a+ neurons are dispensable in adults for PGN-induced PER suggests that for this adult sensory behavior, the ppk23+/Gr66a-neurons that we have demonstrated to respond to PGN (Fig2C-E) are also sufficient (Fig S6A and S6B)."

- The authors do not provide data to show that "Gr66a+/ppk23+ neurons are sufficient" In general the conclusion on the identity of the involved neurons is very indirect and not dissected explicitly. At this stage in our opinion the data the authors show cannot distinguish between "Gr66a+/ppk23+" and "Gr66a+/ppk23-" neurons being important for the analysed behavior.

- lines 654 ff – The panels of Figures S3-S6 are lacking description of what is being shown and the meaning of the used labels.

- Line 852 – "The gut was cut without damaging the brain and taste nerves to allow visual access to the anterior ventral part of the sub-esophageal zone (SEZ)."

o The authors recognize that there is a mistake but did not correct it in the text!

- Our request 2.2

- The authors removed the comparison to MRS without discussing why this was done.

- While the open circles and bars are explained in the statistics section they are still missing in the figure legend which makes it difficult to understand them.

- Our request 2.3

The authors provide new data where they follow our suggestion to directly test their hypothesis by activating Gr66+ neurons through the expression of TRPA1 in axenic larvae. According to their model, this should be sufficient to induce PGN responsive response in adults. The result does not confirm the hypothesis. At this stage they do not discuss the result in a serious way and decide not to include it in the manuscript. We strongly suggest that this is included and that the readers are given the opportunity to see data which do not fully fit the hypothesis. We think this significantly weakens the presented hypothesis but agree that this could also be due to technical challenges of the experiment. This can be discussed but the data should not be omitted. Moreover, could it be that the colour code in the legend is wrong?

- Our request 2.16

- (comment to former line 226) They mention that the time at which they change larvae a new tube is now mentioned in the discussion. This should be also mentioned in the M&M or in the figure legends and not only in the discussion.

As an answer to our last point, the authors write they did not understand the comment.

This was related to the following: In the referred paragraph the authors continue to investigate ppk+/Gr66a- cells. It is not clear to us why, as in the paragraph before they explicitly state that these cells had no behavioral effect: "These data suggest that if Gr66a+ and ppk23+ cells are essential to mediate PGN aversion, the ppk23+/Gr66a- subpopulation is dispensable."

Dear reviewers and editors,

Please find our response to the reviewers' comments

At the end of our responses, we have included the old and the new versions of the main and Suppl figures. In the following text, the reviewers' comments are in black, our response and the original text in green and the new text in red

Reviewer #1 (Remarks to the Author):

The revised version of the manuscript is significantly improved compared to the initial version, the manuscript is relevant and adds interesting new insights into the discussion of the role of bacteria on gustatory mediated behaviors. My major comments have been addressed as well as the minor points have been clarified or corrected. I believe that the new experiments/controls and the more detailed explanation of the different experiments contributed to provide stronger evidence to support the authors' claims.

I have just a few comments that I would like the authors to correct or clarify:

1. I think the reference 38 should be cited again at the end of the sentence in line 33 "PGN signal in these cells", since it is the one that describes the functional requirement of IMD pathway in these cells.

Changes have been made according to reviewer's comment.

Original:

One of them, the membrane receptor PGRP-LC, as well as downstream components of the IMD pathway, are functionally required for transduction of the bacterial PGN signal in these cells^{31, 40, 41}

New version:

One of them, the membrane receptor PGRP-LC, as well as downstream components of the IMD pathway, are functionally required for transduction of the bacterial PGN signal in these cells^{31, 38, 40, 41}

2. I think it is a bit confusing that in the text, line 104-107, the authors refer to the different types of PNG, such as Dap-PNG (PNG100 and PNG 200) and Lys-type PNG but then in figure 1A these compounds are called E.c. 10/100/200 and S.a 100, therefore referring to the bacteria of origin. I think it would be easier to have the same terms to indicate the PNG both in the text and in the Figure.

We changed the text according to the reviewer's comment to homogenize the PGN nomenclature in the figures and in the text.

Original:

we tested whether PGN could suppress the PER response to sucrose. Both Diaminopimelic-type PGN which forms the cell wall of all Gram-negative bacteria and of Bacilli (we used here PGN from Escherichia coli, Ec-PGN), and Lysine-type PGN found in the cell wall of Gram-positive bacteria (we used here PGN from Staphylococcus aureus, Sa-PGN) were tested³³. In PER, DAP-PGN was found to be aversive to reference flies (w-) when used at 100 (PGN100) and 200 µg/mL (PGN200) but not at lower concentrations (Fig 1A). These effects were specific to DAP-type PGN as they were not observed with Lys-type PGN (Fig 1A).

New version:

...we tested whether PGN could suppress the PER response to sucrose. Both Diaminopimelic-type PGN which forms the cell wall of all Gram-negative bacteria and of Bacilli (DAP-PGN; we used here different concentrations of PGN from Escherichia coli, E.c. 10/100/200 µg/mL), and Lysine-type PGN found in the cell wall of Gram-positive bacteria (Lys-PGN; we used here one concentration of PGN from Staphylococcus aureus, S.a. 200 µg/mL) were tested³³. In PER, DAP-PGN was found to be aversive to reference flies (w-) when used at 100 (E.c. 100) and 200 µg/mL (E.c. 200) but not at lower concentrations (Fig 1A). These effects were specific to DAP-type PGN as they were not observed with Lys-type PGN (S.a. 200, Fig 1A)...

3. In line 111 referring to figure S1A/B it is mentioned “sugar or AA rich medium”, but in the figure the protein rich medium is the first represented. It would be better to keep the same order in the text and in the figure and to give the same name to the media (either AA+ or prot+). Same is true for S1A figure legend, line 625.

We changed the text according to the reviewer’s comment to clearly link the text and the order of the experiments presented in the figures.

Original:

Flies reared on both sugar or AA-rich medium displayed an aversion to PGN (Fig S1A and S1B).

New version:

...Flies reared on both protein-rich (prot+) or sugar-rich (suc+) medium displayed an aversion to PGN (Fig S1A and S1B)...

S1A original legend:

(A) and (B), raising medium has no effect on PGN-induced PER suppression. PER index of w- (A) and CantonS (B) flies, raised on sucrose (suc+) or amino acids (prot+) enriched media,

New version:

and (B), raising medium has no effect on PGN-induced PER suppression. PER index of w- (A) and CantonS (B) flies, raised on protein-rich (prot+) or sugar-rich (suc+) media,

4. The paragraph from line 187 and 191 is very important to precisely highlight the conclusions obtained from the experiments aimed at understanding the temporal requirement of the different classes of neurons. I think it should be rephrased more clearly.

The sections 2, 3 and 4 have been rewritten for clarity. Below are the original and new versions.

About Section 2: ***PGN-triggered aversion involves Gr66a+ and ppk23+ neurons***

Original version:

PGN-triggered aversion involves Gr66a+ and ppk23+ neurons

To identify the type of neurons that respond to PGN, PERs were performed in flies in which specific groups of neurons were inactivated by overexpression of the inward rectifier potassium channel Kir2.1. To test whether the bitter network was involved in PER aversion toward DAP-type PGN, neurons expressing the pan-bitter taste receptor Gr66a were inactivated⁴⁷ (Fig S3). The ability of DAP-PGN to suppress PER when mixed with sucrose was completely abolished in Gr66a-Gal4/UAS-Kir2.1 flies demonstrating that Gr66a+ cells are necessary to transduce PGN signal (Fig 2A). Since ppk23+ cells are also found in the taste sensilla and are capable of mediating aversion⁴⁸, we tested their putative implication in mediating PGN aversion (Fig S4). Ppk23-Gal4/UAS-Kir2.1 flies no longer perceived PGN as aversive demonstrating that Gr66a+ and ppk23+ neurons are both necessary to mediate PGN signal (Fig 2B). Recent work has demonstrated the existence of Gr66a+/ppk23+ and Gr66a+/ppk23- neuron sub-populations that are housed in different sensilla^{48, 49}. These two populations display distinct properties related to salt perception. Thus, three subpopulations of taste neurons could exist: those expressing Gr66a but not ppk23 (Gr66a+/ppk23-), those co-expressing Gr66a and ppk23 (Gr66a+/ppk23+) and those expressing ppk23 but not Gr66a (ppk23+/Gr66a). When genetically combined, the ppk23-Gal4 and Gr66a-Gal80 transgenes gave rise to flies in which the Gal4 driver is expressed only in ppk23+/Gr66a- cells⁴⁸. We used this intersectional genetic strategy to specifically visualize and target the ppk23+/Gr66a- cells (Fig S5). Inactivating the ppk23+/Gr66a- population did not impair the adult aversion toward PGN (Fig S6A). These data suggest that if Gr66a+ and ppk23+ cells are essential to mediate PGN aversion, the ppk23+/Gr66a- subpopulation is dispensable. Consequently, the Gr66a+/ppk23+ cells are implicated in the aversion toward PGN (Fig S6A and S6B).

New version:

PGN-triggered aversion involves Gr66a+ and ppk23+ neurons

To identify the type of neurons that respond to PGN, PERs were performed in flies in which specific groups of neurons were inactivated by overexpression of the inward rectifier potassium channel Kir2.1 throughout development (larvae, pupae and adult). To test whether the bitter network was

involved in PER aversion toward DAP-type PGN, neurons producing the pan-bitter taste receptor Gr66a were inactivated⁴⁷ (Fig S3). The ability of DAP-PGN to suppress PER when mixed with sucrose was completely abolished in Gr66a-Gal4/UAS-Kir2.1 flies demonstrating that Gr66a+ cells are necessary to transduce PGN signal (Fig 2A). Another class of neurons able to mediate aversion in the adults has been described in the taste sensilla⁴⁸, they are characterized by the expression of the ppk23 gene (ppk23+) (FigS4). We tested their putative implication in mediating PGN aversion using the Kir2.1 overexpression throughout development. Ppk23-Gal4/UAS-Kir2.1 flies no longer perceived PGN as aversive demonstrating that Gr66a+ and ppk23+ neurons are two cell populations necessary to mediate PGN aversion in the tested adults (Fig 2B). Recent work has shown that flies have 3 neuronal populations expressing either Gr66a and/or ppk23 that display distinct properties related to salt perception^{48,49}: (Gr66a+/ppk23+), (Gr66a+/ppk23-) and (ppk23+/Gr66a-)^{48,49}. In an attempt to delineate whether one of these play a role in the adult aversion process to PGN, we used a genetic intersectional strategy allowing the visualization or functional manipulation of specific subsets on neurons. When genetically combined, with Gal80 being a Gal4 inhibitor, the ppk23-Gal4 and Gr66a-Gal80 transgenes gave rise to flies in which the Gal4 driver is active only in the ppk23+/Gr66a- neuronal subset⁴⁸. We used this intersectional genetic strategy to visualize and target ppk23+/Gr66a- cells (Fig S5). Inactivation of the ppk23+/Gr66a- population alone throughout development using UAS-Kir2.1 did not alter adult PGN aversion (Fig S6A). This negative result indirectly suggests that the remaining cells, i.e. Gr66a+/ppk23- and/or Gr66a+/ppk23+, may be sufficient to mediate PGN aversion (Fig S6A and S6B).

About Section 3: **Adult ppk23+/Gr66a- cells respond to PGN**

Original version:

Adult ppk23+/Gr66a- cells respond to PGN

To demonstrate that PGN can directly activate ppk23+ neurons in the adult, we monitored GCaMP signal in the sub esophageal zone (SEZ) of the brain of females following proboscis exposure to PGN. This brain area processes gustatory input from gustatory neurons located in the proboscis. Since adult Gr66a+ neurons can respond to PGN³⁸, that some labellar Gr66a+ cells are also ppk23+⁴⁸ (Fig S6A and S6B) we used the intersectional genetic strategy described above to quantify calcium activity modifications only in ppk23+/Gr66a- cells⁴⁸. By quantifying live in vivo GCaMP fluorescence in adult brains of flies whose proboscis has been stimulated with PGN, we confirmed that ppk23+/Gr66a- cells can directly respond to PGN (Fig 2C-2E). Importantly, these cells did not respond to caffeine but did to high salt, a signature expected for ppk23+/Gr66a- cells (Fig 2E).

New version:

Adult ppk23+/Gr66a- cells respond to PGN

While functional data point towards a role of Gr66a+ and ppk23+ neuronal groups in mediating adult aversion to PGN, our attempts to define smaller neuronal subpopulations of Gr66a+ or ppk23+ populations were unsuccessful. To have a more direct readout of the effect of PGN on these cells, we monitored calcium level (using GCaMP) in the sub esophageal zone (SEZ) of the brain of females following proboscis exposure to PGN. This brain area processes gustatory input from gustatory neurons located in the proboscis. Our previous work demonstrated that adult Gr66a+ neurons can respond to PGN³⁸. However, as mentioned above, some adult Gr66a+ cells are also ppk23+⁴⁸ (Fig

S6A and S6B). In order to avoid the confusing signal from Gr66a+ cells and to assay whether ppk23+ neurons could on their own respond to PGN, we used the intersectional genetic strategy to quantify calcium activity in ppk23+/Gr66a- cells⁴⁸. Results obtained indicate that ppk23+/Gr66a- cells can directly respond to PGN (Fig 2C-2E). Importantly, these cells did not respond to caffeine but did to high salt, a signature expected for ppk23+/Gr66a- cells (Fig 2E).

About Section 4: **Temporal requirement of gustatory neurons for PGN detection during the fly's lifetime**

Original version:

Temporal requirement of gustatory neurons for PGN detection during the fly's lifetime

The putative involvement of several neuronal subgroups in PGN detection prompted us to test whether these different neurons are all required at the same time during the fly's life. Indeed, although most studies aimed at dissecting how flies perceive their environment via the taste apparatus are performed at the adult stage, including this one, the bitter neurons exemplified by the Gr66a+ population are present throughout fly development^{50, 51}. To determine when the neurons necessary for the PGN-induced PER suppression are required, we took advantage of the Gal4/Gal80^{ts} binary system that allowed us to control the inactivation of taste neurons in a spatially and temporally controlled manner (Fig 3A). Surprisingly, when assessing aversion to PGN, adult flies in which Gr66a+ neurons were functionally inactivated only before the adult stage could no longer perceive PGN as aversive (Fig 3B). In the parallel experiment, inactivation of Gr66a+ neurons in adult flies only, had no effect. In contrast, perception by the adult flies of quinine and caffeine required Gr66a+ neurons to be functional in the adult but not in larvae (Fig S7A). This demonstrates a specific role for Gr66a+ neurons in the life history of the animal, with their requirement only during the larval stage allowing the future adult to show aversion towards DAP-PGN. Since both Gr66a+ and ppk23+ neurons are required to respond to PGN, we tested whether ppk23+ cells are the ones at play in the adults. The data presented Fig 3B demonstrated that, unlike Gr66a+ neurons, ppk23+ neurons are functionally required in the adult, but not during larval life for PGN-triggered aversion. These results reveal an unexpected role for larval Gr66a+ neurons in priming the adult response to PGN.

Other taste receptors have been described and some are co-expressed with Gr66a+, defining subpopulations of Gr66a+ neurons^{43, 52}. In an attempt to map more precisely the subset of Gr66a+ neurons responsible for the phenomenon in larvae, we silenced neurons using drivers known to be co-expressed with Gr66a (Fig 3C). Interestingly, none of the Gr66a+ subpopulations were required for adults to perceive PGN as aversive, indicating that the subset of Gr66a+ required for the phenotype is not defined by co-expression with the drivers we tested (see Discussion). Furthermore, in combination with our data described above indicating that Gr66a+/ppk23+ are sufficient and the demonstration that Gr66a+ neurons are dispensable in adults for PGN-induced PER suggests that for this adult sensory behavior, the ppk23+/Gr66a- neurons that we have demonstrated to respond to PGN (Fig 2C-E) are also sufficient (Fig S6A and S6B).

New version:

Temporal requirement of gustatory neurons for PGN detection during the fly's lifetime

Functional data demonstrate that *Gr66a+* or *ppk23+* neurons inactivation throughout development impairs adult response to PGN. In addition, our previous and current calcium-imaging experiments demonstrate that adult labellar *Gr66a+* neurons as well as *ppk23+/Gr66a-* sub populations respond to PGN. The involvement of several neuronal subgroups in PGN detection prompted us to test whether these different neurons are all required at the same time during the fly's life. Indeed, although most studies aimed at dissecting how flies perceive their environment via the taste apparatus are performed at the adult stage, including this one, the bitter neurons exemplified by the *Gr66a+* population are present throughout fly development^{50, 51}. To determine when the neurons necessary for the PGN-induced PER suppression are required, we took advantage of the *Gal4/Gal80^{ts}* binary system that allowed us to control the inactivation of taste neurons in a spatially and temporally controlled manner (Fig 3A). Surprisingly, when assessing aversion to PGN, adult flies in which *Gr66a+* neurons were functionally inactivated only before the adult stage could no longer perceive PGN as aversive (Fig 3B). In the parallel experiment, inactivation of *Gr66a+* neurons in adult flies only, had no effect. In contrast, perception by the adult flies of quinine and caffeine required *Gr66a+* neurons to be functional in the adult but not in larvae (Fig S7A). Thus, concerning the adult aversion to PGN, *Gr66a+* neurons are not necessary in the adult, but rather during the larval stage. Since both *Gr66a+* and *ppk23+* neurons are required to respond to PGN, we tested whether *ppk23+* cells would be the ones at play in the adults. The data presented Fig 3B demonstrated that, unlike *Gr66a+* neurons, *ppk23+* neurons are functionally required in the adult for the adults PGN-triggered aversion. Furthermore, inactivation of the *ppk23+* neurons only during the larval stage did not impair the PGN-triggered aversion in the adults. These results reveal an unexpected link between larval activity of *Gr66a+* neurons and the adult capacity to respond to PGN that relies on *ppk23+* cells.

Other taste receptors have been described and some are co-expressed with *Gr66a+*, defining subpopulations of *Gr66a+* neurons^{43, 52}. In an attempt to map more precisely the subset of *Gr66a+* neurons responsible for the phenomenon in larvae, we silenced neurons using drivers known to be co-expressed with *Gr66a* (Fig 3C). Our results were negative, with none of the *Gr66a+* subpopulations silenced impairing adult aversion to PGN. These data do not exclude that the tested cells may be involved and indirectly suggest that the subset of *Gr66a+* required for the phenotype may not be defined by co-expression with the drivers we tested (see Discussion). Overall, our data on the temporal inactivation of neuronal groups clarify previous surprising results demonstrating a role for both *Gr66a+* cells and *ppk23+* cells using inactivation throughout development. Indeed, in regards to PGN-triggered aversion in adults, while *Gr66a+* cells are essential during larval life, they are not required at the adult stage. Conversely, *ppk23+* neurons are essential during the adult stage and are not required during larval life. The requirement of a neuronal population being *ppk23+* and not *Gr66a+* (*ppk23+/Gr66a-*) to trigger adult PGN-triggered aversion is in agreement with our *in vivo* calcium imaging assays (Fig 2C-E).

Related to the point about neuronal sub populations, we've made changes in the discussion as follow:

Original:

Since our data show that adult Gr66a+ neurons are dispensable and adult ppk23+ cells essential for PGN-mediated PER, we could conclude that Gr66a-/ppk23+ cells are the only cells involved. However, although Gr66a-/ppk23+ subset can respond specifically to PGN, flies in which Gr66a-/ppk23+ cells are specifically silenced remain able to respond to PGN.

New version:

Since our data show that adult Gr66a+ neurons are dispensable and adult ppk23+ cells are essential for PGN-mediated PER, we suspect that ppk23+/Gr66a- cells are important in the process. However, although the ppk23+/Gr66a- subset may respond to PGN, flies in which ppk23+/Gr66a- cells are silenced remain capable of reacting to the PGN via a reduced PER. One possibility would be that ppk23+/Gr66a+ and ppk23+/Gr66a- neurons could perform the task in adult.

In addition, the following sentence has been added at the end of the discussion section related to **“Are Gr66a+ and ppk23+ subpopulations sufficient to mediate larval priming and adult response to PGN?”**

...All these suggestions are based on experiments that failed to impair the phenotype and are consequently indirect. Further genetic dissection that could provide direct evidences will be necessary to delineate the neuronal sub-populations...

5.The experiment performed to test the involvement of the Toll signaling pathway is not sufficient in my opinion to really rule out its involvement. To get to this conclusion more members of the signaling pathway should be tested but I do not think it is relevant to the current manuscript. If the authors want to keep this experiment in the manuscript, I think they should tune down the interpretation of the results.

We changed the text according to the reviewer’s comment.

Original version:

Since RNAi-mediated downregulation of Myd88 was also without effect (Fig 4C), we ruled out a possible implication of the other fly NF-kB pathway, Toll.

New version:

Since RNAi-mediated downregulation of Myd88, a central component of the Toll pathway did not modify PER response to PGN (Fig 4C), this other NF-kB pathway is unlikely to be involved in adult PGN-triggered aversion. To clearly rule out an implication of the Toll pathway, more elements of the signaling cascade should be tested.

6.In line 258 the sentence “Similarly, rearing germ-free eggs on conventional medium for the first

48 hours of development was sufficient to restore PGN responsiveness in adults (Fig 6D)” does not comment on the 48h-96h interval. In the panel it is evident that the flies coming from the 48h-96h interval show no aversion to PNG or at least significantly reduced aversion compared to the CR flies. However, in the schematic line (blue and gray) on the right it is written 48h-96h AEL aversion to PNG. Can please the author comment on this and rephrase it in the text?

We agree with the reviewer, there was a mistake of the Fig 6D with the wrong indication “Aversion to PGN” while there is No aversion when larvae were exposed to bacteria 48h-96h. This has been fixed in the new Fig 6D and the result has been commented in the main text.

Original version:

Similarly, rearing germ-free eggs on conventional medium for the first 48 hours of development was sufficient to restore PGN responsiveness in adults (Fig 6D).

New version:

Similarly, while rearing larvae from germ-free eggs on conventional medium for the first 48 hours of development was sufficient to restore PGN responsiveness in adults, exposing larvae to conventional medium from 48h to 96h only was not (Fig 6D).

7.I think the data of S2A and S2B should be in the same graph and statistical analysis performed between Sucrose and Sucrose+PNG.

The requested modifications have been made on the figures with the corresponding statistics appearing on the figure and included in the source data file. A version of the figures allowing the reviewer to follow the modifications is provided.

8.In my opinion the genotype of the analyzed flies in the Supplementary figure S3, S4, S5 needs to be indicated in the figure.

The requested modifications have been made on the figures.

9.Referring to the manuscript figures in the discussion session will make the overall discussion much easier to follow.

References to figures that first appear in the main result sections have now been added to illustrate the points mentioned in the discussion.

Reviewer #2 (Remarks to the Author):

Overall there was an effort to provide new data for experiments suggested by the reviewers, such as a thorough analysis of the anatomy of the cells labelled by the used Gal4 lines, testing if the Toll pathway might mediate effects described in the manuscript, and testing if the activation of Gr66+ neurons in larvae are sufficient to induce aversive behaviour in adults. However, many important conclusions remain indirect,

The sections 2, 3 and 4 of the results have been rewritten to take into account the reviewer's comments about the indirect evidences and to clarify the intersectional strategy as well as the conclusions about the neuronal sub populations involved. The changes are detailed below and direct comparison with the original version is provided to allow change tracking.

and many answers were either elusive or several requests for changes in the text were ignored (some even are stated by the authors to have been performed when this was not the case). It was also not trivial to check the edits as not all changes were tracked and especially changes to the figures were not properly documented. This required a lot of extra time and work to be done thoroughly.

For change tracking, the new text appears in red in the main text and a direct comparison allowing Original version vs New version tracking is provided in this letter for all the major modifications and most of the minor ones. In addition, a file allowing the comparison of the previous figures with the new ones is provided.

A main weakness remains that the intersectional genetic dissection of the neuronal substrate of the aversive effect and priming remains very poor and indirect. The authors nevertheless continue making very strong statements.

As described below in more detail, we continue to find their conclusions confusing and often not warranted. Mainly because they are very indirect.

We have rewritten sections 2, 3 and 4 of the results and have added new sentences in the discussion and figure legends to clarify these points. Changes to the text are detailed below. We believe that the clarifications between direct and indirect evidences make the conclusions less puzzling, mainly concerning the neuronal sub-population involved.

Also the statistical tests continue to be problematic and not done across genotypes (with the corresponding controls). While the authors claim they have fixed this or explain why it needs to be done the way they do it, the conclusions they draw requires that they test the effect of for example the suc+Pgn stimulation in the manipulated genotype with the effect of suc+pgn in the control genotypes. This has not been done and it is therefore difficult to make conclusions.

The statistics across genotypes and corresponding conditions were performed, but not added. Now they are present on the figures and the calculations included in the source data file. All the changes to figures are highlighted in a document provided. These statistical analyses did not contradict our

previous conclusions. In the case of the monoassociation with *L. plantarum*, we modulated our conclusions and discussion as detailed below.

Furthermore the authors dismiss many requests to test their hypothesis directly and when they do so (activating specific neurons in axenic larvae (see specific criticism below) they dismiss the results given that they do not provide the expected outcome. They do not discuss why the results are not trustworthy and simply decide to not include the results in the main manuscript. If accepted we strongly advise to include these negative data so that the reader can make their own conclusions about this experiment which fails to support their model. We made specific comments and recommendations below.

The ectopic activation of Gr66a+ neurons during the larval life is now an experiment included in the manuscript and discussed as detailed below (Fig S11B).

Finally, a challenge is that the details of the experimental protocol still remain unclear (e.g. how long was the gustatory stimulation during the imaging experiments etc.).

A clarification to this point has been added to the Mat&Met section. See below.

Original version:

...water on the proboscis. For each condition...

New version in the Mat&Met:

...water on the proboscis. The gustatory stimulation was continuous as the tastant is in the drop contacting the proboscis. The recording started before the addition of the tastant and the calcium response could be observed immediately following the contact with the sensilla. For each condition...

In general the phenomenology is interesting and worthy of a thorough mechanistic dissection. The main results describing the phenomenology, should be published as they are valuable for the community. But the current conclusions are very strongly worded and the mechanistic dissection remains rather superficial, indirect, and not very well supported or contradicted by specific results. The revision is a missed opportunity and in our opinion the manuscript should not be published in its current form.

Morespecifically:

- line 127 - This whole section is (still) very confusing. I think it would help the reader if the authors could explicitly state here, that the manipulations made are effective throughout the life of the animal. for example, by adding something like this:

- "To identify the type of neurons that respond to PGN, PER assays were performed in flies in which 127 specific groups of neurons were inactivated by overexpression of the inward rectifier potassium channel Kir2.1 throughout development."

The suggested modification has been made and clarifications appear in different parts of the text (in red) including elements in the legend of Figure S6 that explains the sub-populations and the intersectional strategy.

In addition, this important section about Gr66a+ cells and ppk23+ cells involvement has been modified following reviewer's comments and suggestions to clearly discriminate suggestions from demonstrations and direct evidences from indirect ones.

About Section 2: **PGN-triggered aversion involves Gr66a+ and ppk23+ neurons**

Original version:

PGN-triggered aversion involves Gr66a+ and ppk23+ neurons

To identify the type of neurons that respond to PGN, PERs were performed in flies in which specific groups of neurons were inactivated by overexpression of the inward rectifier potassium channel Kir2.1. To test whether the bitter network was involved in PER aversion toward DAP-type PGN, neurons expressing the pan-bitter taste receptor Gr66a were inactivated⁴⁷ (Fig S3). The ability of DAP-PGN to suppress PER when mixed with sucrose was completely abolished in Gr66a-Gal4/UAS-Kir2.1 flies demonstrating that Gr66a+ cells are necessary to transduce PGN signal (Fig 2A). Since ppk23+ cells are also found in the taste sensilla and are capable of mediating aversion⁴⁸, we tested their putative implication in mediating PGN aversion (Fig S4). Ppk23-Gal4/UAS-Kir2.1 flies no longer perceived PGN as aversive demonstrating that Gr66a+ and ppk23+ neurons are both necessary to mediate PGN signal (Fig 2B). Recent work has demonstrated the existence of Gr66a+/ppk23+ and Gr66a+/ppk23- neuron sub-populations that are housed in different sensilla^{48, 49}. These two populations display distinct properties related to salt perception. Thus, three subpopulations of taste neurons could exist: those expressing Gr66a but not ppk23 (Gr66a+/ppk23-), those co-expressing Gr66a and ppk23 (Gr66a+/ppk23+) and those expressing ppk23 but not Gr66a (ppk23+/Gr66a). When genetically combined, the ppk23-Gal4 and GR66a-Gal80 transgenes gave rise to flies in which the Gal4 driver is expressed only in ppk23+/Gr66a- cells⁴⁸. We used this intersectional genetic strategy to specifically visualize and target the ppk23+/Gr66a- cells (Fig S5). Inactivating the ppk23+/Gr66a- population did not impair the adult aversion toward PGN (Fig S6A). These data suggest that if Gr66a+ and ppk23+ cells are essential to mediate PGN aversion, the ppk23+/Gr66a- subpopulation is dispensable. Consequently, the Gr66a+/ppk23+ cells are implicated in the aversion toward PGN (Fig S6A and S6B).

New version:

PGN-triggered aversion involves Gr66a+ and ppk23+ neurons

To identify the type of neurons that respond to PGN, PERs were performed in flies in which specific groups of neurons were inactivated by overexpression of the inward rectifier potassium channel Kir2.1 throughout development (larvae, pupae and adult). To test whether the bitter network was involved in PER aversion toward DAP-type PGN, neurons producing the pan-bitter taste receptor Gr66a were inactivated⁴⁷ (Fig S3). The ability of DAP-PGN to suppress PER when mixed with sucrose was completely abolished in Gr66a-Gal4/UAS-Kir2.1 flies demonstrating that Gr66a+ cells are necessary to transduce PGN signal (Fig 2A). Another class of neurons able to mediate aversion in the adults has been described in the taste sensilla⁴⁸, they are characterized by the expression of the ppk23 gene (ppk23+) (FigS4). We tested their putative implication in mediating PGN aversion using the Kir2.1 overexpression throughout development. Ppk23-Gal4/UAS-Kir2.1 flies no longer perceived

PGN as aversive demonstrating that Gr66a+ and ppk23+ neurons are two cell populations necessary to mediate PGN aversion in the tested adults (Fig 2B). Recent work has shown that flies have 3 neuronal populations expressing either Gr66a and/or ppk23 that display distinct properties related to salt perception^{48,49}: (Gr66a+/ppk23+), (Gr66a+/ppk23-) and (ppk23+/Gr66a-)^{48,49}. In an attempt to delineate whether one of these play a role in the adult aversion process to PGN, we used a genetic intersectional strategy allowing the visualization or functional manipulation of specific subsets on neurons. When genetically combined, with Gal80 being a Gal4 inhibitor, the ppk23-Gal4 and Gr66a-Gal80 transgenes gave rise to flies in which the Gal4 driver is active only in the ppk23+/Gr66a- neuronal subset⁴⁸. We used this intersectional genetic strategy to visualize and target ppk23+/Gr66a- cells (Fig S5). Inactivation of the ppk23+/Gr66a- population alone throughout development using UAS-Kir2.1 did not alter adult PGN aversion (Fig S6A). This negative result indirectly suggests that the remaining cells, i.e. Gr66a+/ppk23- and/or Gr66a+/ppk23+, may be sufficient to mediate PGN aversion (Fig S6A and S6B).

- line 141- "(ppk23+/Gr66a)" a - is missing form (ppk23+/Gr66a*-*).

This has been changed.

- line 145 - "Inactivating the ppk23+/Gr66a- population did not impair the adult aversion toward PGN (Fig S6A). These data suggest that if Gr66a+ and ppk23+ cells are essential to mediate PGN aversion, the ppk23+/Gr66a- subpopulation is dispensable. Consequently, the Gr66a+/ppk23+ cells are implicated in the aversion toward PGN (Fig S6A and S6B)."

- The authors cannot exclude that Gr66a+/ppk23- play a role here as well and should state this here.

Following previous comments by the reviewer, this pivotal section that was still confusing has been modified following reviewer's comments and suggestions to clearly discriminate suggestions from demonstrations and direct evidences from indirect ones.

The new version is presented above.

- line 154 - a word seems missing in the sentence:

- "Since adult Gr66a+ neurons can respond to PGN, that some labellar Gr66a+ cells are also ppk23+ (Fig S6A and S6B) we used the intersectional genetic strategy described above to quantify calcium activity modifications only in ppk23+/Gr66a- cells 48."

This comment has been addressed in the whole section changes described below.

• line 154 – It is not clear what is the logic and relevance of imaging ppk23+/Gr66a- neurons if these neurons were not necessary for behavior? Also not how this finding is relevant for the conclusions. This important section that was still lacking clarifications has been modified following reviewer's comments and suggestions to explain the logic and relevance of the approach involving calcium imaging and ppk23+/Gr66a- cells.

About Section 3: **Adult ppk23+/Gr66a- cells respond to PGN**

Original version:

Adult ppk23+/Gr66a- cells respond to PGN

To demonstrate that PGN can directly activate ppk23+ neurons in the adult, we monitored GCaMP signal in the sub esophageal zone (SEZ) of the brain of females following proboscis exposure to PGN. This brain area processes gustatory input from gustatory neurons located in the proboscis. Since adult Gr66a+ neurons can respond to PGN³⁸, that some labellar Gr66a+ cells are also ppk23+⁴⁸ (Fig S6A and S6B) we used the intersectional genetic strategy described above to quantify calcium activity modifications only in ppk23+/Gr66a- cells⁴⁸. By quantifying live in vivo GCaMP fluorescence in adult brains of flies whose proboscis has been stimulated with PGN, we confirmed that ppk23+/Gr66a- cells can directly respond to PGN (Fig 2C-2E). Importantly, these cells did not respond to caffeine but did to high salt, a signature expected for ppk23+/Gr66a- cells (Fig 2E).

New version:

Adult ppk23+/Gr66a- cells respond to PGN

While functional data point towards a role of Gr66a+ and ppk23+ neuronal groups in mediating adult aversion to PGN, our attempts to define smaller neuronal subpopulations of Gr66a+ or ppk23+ populations were unsuccessful. To have a more direct readout of the effect of PGN on these cells, we monitored calcium level (using GCaMP) in the sub esophageal zone (SEZ) of the brain of females following proboscis exposure to PGN. This brain area processes gustatory input from gustatory neurons located in the proboscis. Our previous work demonstrated that adult Gr66a+ neurons can respond to PGN³⁸. However, as mentioned above, some adult Gr66a+ cells are also ppk23+⁴⁸ (Fig S6A and S6B). In order to avoid the confusing signal from Gr66a+ cells and to assay whether ppk23+ neurons could on their own respond to PGN, we used the intersectional genetic strategy to quantify calcium activity in ppk23+/Gr66a- cells⁴⁸. Results obtained indicate that ppk23+/Gr66a- cells can directly respond to PGN (Fig 2C-2E). Importantly, these cells did not respond to caffeine but did to high salt, a signature expected for ppk23+/Gr66a- cells (Fig 2E).

- line 187ff - "Furthermore, in combination with our data described above indicating that Gr66a+/ppk23+ are sufficient and the demonstration that Gr66a+ neurons are dispensable in adults for PGN-induced PER suggests that for this adult sensory behavior, the ppk23+/Gr66a-neurons that we have demonstrated to respond to PGN (Fig2C-E) are also sufficient (Fig S6A and S6B)."
- The authors do not provide data to show that "Gr66a+/ppk23+ neurons are sufficient" In general the conclusion on the identity of the involved neurons is very indirect and not dissected explicitly. At this stage in our opinion the data the authors show cannot distinguish between "Gr66a+/ppk23+" and "Gr66a+/ppk23-" neurons being important for the analysed behavior.

This crucial section about the temporal requirement of the neuronal groups required further clarifications. The whole section has been modified following reviewer's comments and suggestions in order to explain the logic of the approach as well as to draw conclusions only from direct evidences.

About Section 4: **Temporal requirement of gustatory neurons for PGN detection during the fly's lifetime**

Original version:

Temporal requirement of gustatory neurons for PGN detection during the fly's lifetime

The putative involvement of several neuronal subgroups in PGN detection prompted us to test whether these different neurons are all required at the same time during the fly's life. Indeed, although most studies aimed at dissecting how flies perceive their environment via the taste apparatus are performed at the adult stage, including this one, the bitter neurons exemplified by the Gr66a+ population are present throughout fly development^{50, 51}. To determine when the neurons necessary for the PGN-induced PER suppression are required, we took advantage of the Gal4/Gal80^{ts} binary system that allowed us to control the inactivation of taste neurons in a spatially and temporally controlled manner (Fig 3A). Surprisingly, when assessing aversion to PGN, adult flies in which Gr66a+ neurons were functionally inactivated only before the adult stage could no longer perceive PGN as aversive (Fig 3B). In the parallel experiment, inactivation of Gr66a+ neurons in adult flies only, had no effect. In contrast, perception by the adult flies of quinine and caffeine required Gr66a+ neurons to be functional in the adult but not in larvae (Fig S7A). This demonstrates a specific role for Gr66a+ neurons in the life history of the animal, with their requirement only during the larval stage allowing the future adult to show aversion towards DAP-PGN. Since both Gr66a+ and ppk23+ neurons are required to respond to PGN, we tested whether ppk23+ cells are the ones at play in the adults. The data presented Fig 3B demonstrated that, unlike GR66a+ neurons, ppk23+ neurons are functionally required in the adult, but not during larval life for PGN-triggered aversion. These results reveal an unexpected role for larval Gr66a+ neurons in priming the adult response to PGN.

Other taste receptors have been described and some are co-expressed with Gr66a+, defining subpopulations of Gr66a+ neurons^{43, 52}. In an attempt to map more precisely the subset of Gr66a+ neurons responsible for the phenomenon in larvae, we silenced neurons using drivers known to be co-expressed with Gr66a (Fig 3C). Interestingly, none of the Gr66a+ subpopulations were required for adults to perceive PGN as aversive, indicating that the subset of Gr66a+ required for the phenotype is not defined by co-expression with the drivers we tested (see Discussion). Furthermore, in combination with our data described above indicating that Gr66a+/ppk23+ are sufficient and the demonstration that Gr66a+ neurons are dispensable in adults for PGN-induced PER suggests that for this adult sensory behavior, the ppk23+/Gr66a- neurons that we have demonstrated to respond to PGN (Fig2C-E) are also sufficient (Fig S6A and S6B).

New version:

Temporal requirement of gustatory neurons for PGN detection during the fly's lifetime

Functional data demonstrate that Gr66a+ or ppk23+ neurons inactivation throughout development impairs adult response to PGN. In addition, our previous and current calcium-imaging experiments demonstrate that adult labellar Gr66a+ neurons as well as ppk23+/Gr66a- sub populations respond to PGN. The involvement of several neuronal subgroups in PGN detection prompted us to test whether these different neurons are all required at the same time during the fly's life. Indeed, although most studies aimed at dissecting how flies perceive their environment via the taste apparatus are performed at the adult stage, including this one, the bitter neurons exemplified by the Gr66a+ population are present throughout fly development^{50, 51}. To determine when the neurons necessary for the PGN-induced PER suppression are required, we took advantage of the Gal4/Gal80^{ts} binary system that allowed us to control the inactivation of taste neurons in a spatially and temporally controlled manner (Fig 3A). Surprisingly, when assessing aversion to PGN, adult flies in which Gr66a+ neurons were functionally inactivated only before the adult stage could no longer perceive PGN as aversive (Fig 3B). In the parallel experiment, inactivation of Gr66a+ neurons in adult flies only, had no effect. In contrast, perception by the adult flies of quinine and caffeine required Gr66a+ neurons to be functional in the adult but not in larvae (Fig S7A). Thus, concerning the adult aversion to PGN, Gr66a+ neurons are not necessary in the adult, but rather during the larval stage. Since both Gr66a+ and ppk23+ neurons are required to respond to PGN, we tested whether ppk23+ cells would be the ones at play in the adults. The data presented Fig 3B demonstrated that, unlike Gr66a+ neurons, ppk23+ neurons are functionally required in the adult for the adults PGN-triggered aversion. Furthermore, inactivation of the ppk23+ neurons only during the larval stage did not impair the PGN-triggered aversion in the adults. These results reveal an unexpected link between larval activity of Gr66a+ neurons and the adult capacity to respond to PGN that relies on ppk23+ cells.

Other taste receptors have been described and some are co-expressed with Gr66a+, defining subpopulations of Gr66a+ neurons^{43, 52}. In an attempt to map more precisely the subset of Gr66a+ neurons responsible for the phenomenon in larvae, we silenced neurons using drivers known to be co-expressed with Gr66a (Fig 3C). Our results were negative, with none of the Gr66a+ subpopulations silenced impairing adult aversion to PGN. These data do not exclude that the tested cells may be involved and indirectly suggest that the subset of Gr66a+ required for the phenotype may not be defined by co-expression with the drivers we tested (see Discussion). Overall, our data on the temporal inactivation of neuronal groups clarify previous surprising results demonstrating a role for both Gr66a+ cells and ppk23+ cells using inactivation throughout development. Indeed, in regards to PGN-triggered aversion in adults, while Gr66a+ cells are essential during larval life, they are not required at the adult stage. Conversely, ppk23+ neurons are essential during the adult stage and are not required during larval life. The requirement of a neuronal population being ppk23+ and not Gr66a+ (ppk23+/Gr66a-) to trigger adult PGN-triggered aversion is in agreement with our in vivo calcium imaging assays (Fig 2C-E).

Related to the point about neuronal sub populations, we've made changes in the discussion "**Are Gr66a+ and ppk23+ subpopulations sufficient to mediate larval priming and adult response to PGN?**" as follow:

Original:

Since our data show that adult Gr66a+ neurons are dispensable and adult ppk23+ cells essential for PGN-mediated PER, we could conclude that Gr66a-/ppk23+ cells are the only cells involved. However,

although Gr66a-/ppk23+ subset can respond specifically to PGN, flies in which Gr66a-/ppk23+ cells are specifically silenced remain able to respond to PGN.

New version:

Since our data show that adult Gr66a+ neurons are dispensable and adult ppk23+ cells are essential for PGN-mediated PER, we suspect that ppk23+/Gr66a- cells are important in the process. However, although the ppk23+/Gr66a- subset may respond to PGN, flies in which ppk23+/Gr66a- cells are silenced remain capable of reacting to the PGN via a reduced PER. One possibility would be that ppk23+/Gr66a+ and ppk23+/Gr66a- neurons could perform the task in adult.

In addition, the following sentence has been added at the end of the discussion section related to **“Are Gr66a+ and ppk23+ subpopulations sufficient to mediate larval priming and adult response to PGN?”**

...All these suggestions are based on experiments that failed to impair the phenotype and are consequently indirect. Further genetic dissection that could provide direct evidences will be necessary to delineate the neuronal sub-populations...

- lines 654 ff – The panels of Figures S3-S6 are lacking description of what is being shown and the meaning of the used labels.

Legends for S3, S4 and S5 have been expanded to introduce what is shown and the labels are now explained.

Original version:

Fig.S3

Expression pattern of the Gr66a-Gal4 driver. Gr66a-Gal4/UAS-GFP larvae and adult females were dissected and the fluorescence was observed in larval taste neurons, projections in larval brain, adult taste neurons of the proboscis, projections in adult brains and terminal tarsi of the legs. We did not observe any fluorescence in larval gut, larval fat body, adult gut, adult carcass or adult ovaries.

Fig. S4

Expression pattern of the ppk23-Gal4 driver. Ppk23-Gal4/UAS-GFP larvae and adult females were dissected and the fluorescence was observed in larval taste neurons, projections in larval brain, adult taste neurons of the proboscis, projections in adult brains and terminal tarsi of the legs. We did not observe any fluorescence in larval gut, larval fat body, adult gut, adult carcass or adult ovaries.

Fig. S5

Expression pattern of the ppk23-Gal4 driver combined to Gr66a-LexA; LexAop-Gal80 using UAS-GFP. Larvae and adult females were dissected and the fluorescence was observed in larval taste neurons, projections in larval brain, adult taste neurons of the proboscis, projections in adult brains and terminal tarsi of the legs. We did not observe any fluorescence in larval gut, larval fat body, adult gut, adult carcass or adult ovaries.

New version following reviewer's comments:

Fig.S3

*Expression pattern of the Gr66a-Gal4 driver. Gr66a-Gal4/UAS-GFP larvae and adult females were dissected and the fluorescence was observed in larval taste neurons (first line pictures), projections in larval brain, adult taste neurons of the proboscis, projections in adult brains (third line pictures) and terminal tarsi of the legs (fifth line picture). We did not observe any fluorescence in larval gut (second line pictures), larval fat body (second line pictures), adult gut, adult carcass or adult ovaries (fourth line pictures). TO for Terminal Organ; DO for Dorsal Organ; TOG for Terminal Organ Ganglion; DOG for Dorsal Organ Ganglion; DPS for Dorsal Pharyngeal Sensory organ; VPS for Ventral Pharyngeal Sensory organ. P for axons emanating from neurons in the Proboscis; PH for axons emanating from neurons in the Pharynx; L for axons emanating from neurons in the Legs. MP for Maxillary Palp; LP for Labellum-Proboscis. The * defines the anterior of the portion shown. The boxes within pictures are magnifications of the area delimited by the dashed line. D, V, P and A for Dorsal, Ventral, Posterior and Anterior.*

Fig. S4

*Expression pattern of the ppk23-Gal4 driver. Ppk23-Gal4/UAS-GFP larvae and adult females were dissected and the fluorescence was observed in larval taste neurons, (first line pictures), projections in larval brain, adult taste neurons of the proboscis, projections in adult brains (third line pictures) and terminal tarsi of the legs (fifth line picture). We did not observe any fluorescence in larval gut (second line pictures), larval fat body (second line pictures), adult gut, adult carcass or adult ovaries (fourth line pictures). TO for Terminal Organ; TOG for Terminal Organ Ganglion; DOG for Dorsal Organ Ganglion; DPS for Dorsal Pharyngeal Sensory organ; VPS for Ventral Pharyngeal Sensory organ. P for axons emanating from neurons in the Proboscis. LP for Labellum-Proboscis. The * defines the anterior of the portion shown. The boxes within pictures are magnifications of the area delimited by the dashed line. D, V, P and A for Dorsal, Ventral, Posterior and Anterior.*

Fig. S5

Expression pattern of the ppk23-Gal4 driver combined to Gr66a-LexA; LexAop-Gal80 using UAS-GFP. Larvae and adult females were dissected and the fluorescence was observed in larval taste neurons, (first line pictures), projections in larval brain, adult taste neurons of the proboscis, projections in

adult brains (third line pictures) and terminal tarsi of the legs (fifth line picture). We did not observe any fluorescence in larval gut (second line pictures), larval fat body (second line pictures), adult gut, adult carcass or adult ovaries (fourth line pictures). TO for Terminal Organ; TOG for Terminal Organ Ganglion; DOG for Dorsal Organ Ganglion. P for axons emanating from neurons in the Proboscis; L for axons emanating from neurons in the Legs. LP for Labellum-Proboscis. The * defines the anterior of the portion shown. The boxes within pictures are magnifications of the area delimited by the dashed line. D, V, P and A for Dorsal, Ventral, Posterior and Anterior.

- Line 852 – “The gut was cut without damaging the brain and taste nerves to allow visual access to the anterior ventral part of the sub-esophageal zone (SEZ).”
 - o The authors recognize that there is a mistake but did not correct it in the text!

The requested changes have been included in the new version.

Original version:

the fat body were removed. The gut was cut without damaging the brain

New version:

the fat body was removed. The esophagus was cut without damaging the brain

- Our request 2.2

- The authors removed the comparison to MRS without discussing why this was done.

Following reviewer’s comments about this section, we added the statistics across conditions and genotypes. New statistics now appear on the figures and the data are in the source data file. All the new statistics present in the figures are highlighted in the Figure Comparison Document. Specifically, about *L. plantarum* monoassociation, our conclusion was modulated as follow.

Original version:

When the same experiment was performed with a related species, *Lactobacillus plantarum* (*L. plantarum*), no PER inhibition was observed (Fig 7C).

New version:

When the same experiment was performed with a related species, *Lactobacillus plantarum* (*L. plantarum*), there was a trend toward PER inhibition, but not as robust as the one we quantified using mono-association with *L. brevis* (Fig 7C).

And in the discussion:

While the cohabitation of larvae with a bacteria considered as pathobiont like *L. brevis*⁶⁸, makes the adults from which they are produced wary of PGN (Fig 7C), this is less the case for the commensal bacterium *L. plantarum*.

- While the open circles and bars are explained in the statistics section they are still missing in the figure legend which makes it difficult to understand them.

We added an explanation of our PER data representation adapted from the one present in the Mat & Met in the legend of the Figure 1 and we expect that this full description of the first PER graphic representation will be sufficient for the remaining figures.

Original version of the legend for the Fig 1:

Fig. 1

*Dose-dependent PGN inhibition of PER. (A) PER index of w- flies to control solutions of sucrose 1mM and sucrose 1mM + caffeine 10mM and to sucrose 1mM + increasing concentrations of PGN from E. coli K12 (Ec) and S. aureus (SA). The numbers below the x-axis correspond to the PGN final concentrations in µg/mL. (B and C) Aversion to PGN is independent of genetic background. PER index of yw- (B) and CantonS (C) flies to control solutions of sucrose 1mM and sucrose 1mM + caffeine 10mM and to sucrose 1mM + PGN from E. coli K12 at 200µg/mL. The PER index is calculated as the percentage of flies tested that responded with a PER to the stimulation ± 95% confidence interval (CI). A PER value of 1 means that 100% of the tested flies extended their proboscis following contact with the mixture, a value of 0,2 means that 20% of the animals extended their proboscis. The number of tested flies (n) is indicated on top of each bar. ns indicates p>0.05, * indicates p<0.05, ** indicates p<0.01, *** indicates p<0.001, **** indicates p<0.0001 Fisher Exact Test. Further details can be found in the detailed lines, conditions and, statistics for the figure section.*

New version of the legend for the Fig 1:

Fig. 1

Dose-dependent PGN inhibition of PER. (A) PER index of w- flies to control solutions of sucrose 1mM and sucrose 1mM + caffeine 10mM and to sucrose 1mM + increasing concentrations of PGN from E. coli K12 (Ec) and S. aureus (SA). The numbers below the x-axis correspond to the PGN final concentrations in µg/mL. (B and C) Aversion to PGN is independent of genetic background. PER index of yw- (B) and CantonS (C) flies to control solutions of sucrose 1mM and sucrose 1mM + caffeine 10mM and to sucrose 1mM + PGN from E. coli K12 at 200µg/mL. The PER index is calculated as the percentage of flies tested that responded with a PER to the stimulation ± 95% confidence interval (CI). The total number of tested flies (n) for one condition is indicated on top of each bar. As the values obtained from one fly are categorical data with a Yes or No value, we used the Fisher exact t-test and the 95% CI to test the statistical significance of a possible difference between a test sample and the related control. At least 3 independent experiments were performed. The results from all the

*independent experiments were gathered and consequently, we do not show the average response from one experiment representative of the different biological replicates, but an average from all the data generated during the independent experiments in one graph. However, each open circle represents the average PER of one experiment. A PER value of 1 means that 100% of the tested flies extended their proboscis following contact with the mixture, a value of 0,2 means that 20% of the animals extended their proboscis. The number of tested flies (n) is indicated on top of each bar. ns indicates $p>0.05$, * indicates $p<0.05$, ** indicates $p<0.01$, *** indicates $p<0.001$, **** indicates $p<0.0001$ Fisher Exact Test. Further details can be found in the source data file.*

- Our request 2.3

The authors provide new data where they follow our suggestion to directly test their hypothesis by activating Gr66+ neurons through the expression of TRPA1 in axenic larvae. According to their model, this should be sufficient to induce PGN responsive response in adults.

The result does not confirm the hypothesis. At this stage they do not discuss the result in a serious way and decide not to include it in the manuscript. We strongly suggest that this is included and that the readers are given the opportunity to see data which do not fully fit the hypothesis. We think this significantly weakens the presented hypothesis but agree that this could also be due to technical challenges of the experiment. This can be discussed but the data should not be omitted. Moreover, could it be that the colour code in the legend is wrong?

This is always an issue to include or not a negative result. We have off course no problem to include it. There are two ways to analyze this result. Either the mode of activation (duration, intensity...) of the neuron does not mimic the one that takes place *in vivo*, or the activation of Gr66a cells *per se* is not sufficient to trigger the phenotype.

The new data are now included as Fig S11B, introduced, described and commented as follow:
We got rid of the color code to clearly explain the ectopic activation.

Based on our data, a simple model would be that activation of Gr66a+ during larval life, directly or indirectly linked to cohabitation with bacteria, is sufficient to generate adults with the ability to avoid PGN. To test this hypothesis, we ectopically activated the Gr66a+ neurons during larval life only using over-expression of a heat-sensitive TRPA1 isoform. Expression of this construct in neurons and exposition of animals to temperatures above 23°C trigger a calcium influx that can activate the neuron despite the absence of endogenous stimulation. Thus, constitutive activation of Gr66a+ neurons in germ-free larvae may be sufficient to obtain PGN responsive adults. We performed the test with germ-free larvae and observe no aversion to PGN in the resulting adults (Fig S11B). Thus, activation of Gr66a+ neurons under our conditions is not sufficient. The type and intensity of activation that we used may not mimic the physiological ones. Alternatively, the system may require

multiple inputs, the activation of Gr66a+ larval cells being only one of them.

New Fig Legend

(B) Activating ectopically Gr66a+ neurons during the larval life in germ-free (GF) conditions is not sufficient to obtain adults reacting to PGN. PER index of animals with the UAS-TRPA1^{ts} construction under the control of the Gr66a-Gal4 driver to control sucrose solution and to sucrose 1mM + PGN from *E. coli* K12 at 200µg/mL. Larvae and adults were raised at 18°C on classical media (CR) or germ-free media (GF) or on germ-free media with larvae at 29°C and adults at 18°C so the ectopic activation of Gr66a+ neurons is only triggered during the larval life.

- Our request 2.16

- (comment to former line 226) They mention that the time at which they change larvae a new tube is now mentioned in the discussion. This should be also mentioned in the M&M or in the figure legends and not only in the discussion.

In the mentioned experiments, germ-free condition was maintained throughout development. In the next section however, stage-specific germ-free conditions are used. Therefore, we modified the text of the first section to insist on the treatment all lifelong.

Original:

...When flies were reared on antibiotic (germ-free) medium, ...

New version:

...When animals were reared on antibiotic (germ-free) medium throughout development (larval and adult life), ...

Moreover, we modified the Mat et Met section related to Larvae contamination, as shown below

Original version

Larvae contamination

An oviposition of w- germ free flies is set up on an apple agar plate at 25°C for 4h. Simultaneously an oviposition of yw- flies is set up in a regular fly tube. After 4h the w- embryos are sterilized (see protocol above) and transferred on antibiotic enriched media. Once they reach the desired developmental stage w- larvae are filtered out of the antibiotic enriched media and rinsed off with PBS. They are then transferred in the tube in which yw- larvae are growing. In this way w- larvae are exposed to the same bacteria as yw- flies but starting at a defined developmental stage. Once the adults emerge from the pupae, they are distinguished on the basis of body color. yw- flies were

exposed to the bacteria throughout their lives, whereas w- flies were only exposed from the moment of transfer.

New version:

Larvae contamination

An oviposition of w- germ free flies is set up on an apple agar plate at 25°C for 4h. Simultaneously an oviposition of yw- flies is set up in a regular fly tube. After 4h the w- embryos are sterilized (see protocol above) and transferred on antibiotic enriched media. Once they reach the desired developmental stage w- larvae are filtered out of the antibiotic enriched media and rinsed off with PBS. They are then transferred in the tube in which yw- larvae are growing. In this way w- larvae are exposed to the same bacteria as yw- flies but starting at a defined developmental stage. Once the adults emerge from the pupae, they can be transferred to germ-free tubes and they are distinguished on the basis of body color. This way, while yw- animals were exposed to the bacteria throughout their lives, w- animals are exposed in a controlled manner depending on the moment of transfer.

As an answer to our last point, the authors write they did not understand the comment. This was related to the following: In the referred paragraph the authors continue to investigate ppk+/Gr66a- cells. It is not clear to us why, as in the paragraph before they explicitly state that these cells had no behavioral effect: "These data suggest that if Gr66a+ and ppk23+ cells are essential to mediate PGN aversion, the ppk23+/Gr66a- subpopulation is dispensable."

Indeed, the sentences dealing with the different subsets were confusing and even puzzling. We believe that the rewriting of the sections 2, 3 and 4 that we propose in the revised version clarifies it.

FIGURE 1 OLD VERSION

A

B

C

FIGURE 1 NEW VERSION

A

B

C

FIGURE 2 OLD VERSION

FIGURE 2 NEW VERSION

FIGURE 3 OLD VERSION

A

B

C

FIGURE 3 NEW VERSION

A

B

C

FIGURE 4 OLD VERSION

FIGURE 4 NEW VERSION

FIGURE 5 OLD VERSION

A

B

C

D

E

FIGURE 5 NEW VERSION

A

B

C

D

E

FIGURE 6 OLD VERSION

A

B

C

D

FIGURE 6 NEW VERSION

A

B

C

D

FIGURE 7 OLD VERSION

A

B

C

D

E

F

FIGURE 7 NEW VERSION

A

B

C

D

E

F

FIGURE S1 OLD VERSION

A

B

C

FIGURE S1 NEW VERSION

A

B

C

FIGURE S2 OLD VERSION

A

B

C

FIGURE S2 NEW VERSION

A

B

FIGURE S3 OLD VERSION

A

A**FIGURE S3** NEW VERSION

Gr66a-Gal4/UAS-GFP

FIGURE S4 OLD VERSION

A

A**FIGURE S4** NEW VERSION

ppk23-Gal4/UAS-GFP

FIGURE S5 OLD VERSION

A

FIGURE S5 NEW VERSION

Gr66a-LexA;LexAop-Gal80;ppk23-Gal4/UAS-GFP

A

FIGURE S6 OLD VERSION

A

B

FIGURE S6 NEW VERSION

A

B

FIGURE S7 OLD VERSION

A

18°C permissive 29°C restrictive

FIGURE S7 NEW VERSION

A

18°C permissive 29°C restrictive

FIGURE S8 OLD VERSION

A

B

C

FIGURE S8 NEW VERSION

A

B

C

FIGURE S9 OLD VERSION

FIGURE S9 NEW VERSION

FIGURE S10 OLD VERSION

A

Conventionally raised(CR)

Germ-free (GF)

B

C

FIGURE S10 NEW VERSION

FIGURE S11 OLD VERSION

A

B

FIGURE S11 NEW VERSION

A

B

C

REVIEWERS' COMMENTS

Reviewer #1 (Remarks to the Author):

As mentioned after the previous round of revision, the latest version of the manuscript is significantly improved compared to the initial version. In particular, I had only some minor comments and the authors addressed them all and clarified the points that I thought needed to be addressed.